# Immunopeptidomics-based identification of naturally presented non-canonical circRNA-derived peptides

Humberto J. Ferreira [1,2,3], Brian J. Stevenson[1,3,4], HuiSong Pak[1,2,3], Fengchao Yu [5], Jessica Almeida Oliveira [1,2,3], Florian Huber [1,2,3], Marie Taillandier-Coindard[1,2,3], Justine Michaux[1,2,3], Emma Ricart-Altimiras[1,2,3], Anne I. Kraemer[1,2,3], Lana E. Kandalaft [1,2,3,6], Daniel E. Speiser [2], Alexey I. Nesvizhskii [5,7], Markus Müller [1,2,3,4] & Michal Bassani-Sternberg [1,2,3,6] ✉

Circular RNAs (circRNAs) are covalently closed non-coding RNAs lacking the 5' cap and the poly-A tail. Nevertheless, it has been demonstrated that certain circRNAs can undergo active translation. Therefore, aberrantly expressed circRNAs in human cancers could be an unexplored source of tumor-specific antigens, potentially mediating anti-tumor T cell responses. This study presents an immunopeptidomics workflow with a specific focus on generating a circRNA-specific protein fasta reference. The main goal of this workflow is to streamline the process of identifying and validating human leukocyte antigen (HLA) bound peptides potentially originating from circRNAs. We increase the analytical stringency of our workflow by retaining peptides identified independently by two mass spectrometry search engines and/or by applying a group-specific FDR for canonical-derived and circRNA-derived peptides. A subset of circRNA-derived peptides specifically encoded by the region spanning the back-splice junction (BSJ) are validated with targeted MS, and with direct Sanger sequencing of the respective source transcripts. Our workflow identifies 54 unique BSJ-spanning circRNA-derived peptides in the immunopeptidome of melanoma and lung cancer samples. Our approach enlarges the catalog of source proteins that can be explored for immunotherapy.

Adoptive T cell-based immunotherapies and cancer vaccines are becoming powerful cancer treatment options[1]. They leverage natural anti-cancer immunity by targeting human leukocyte antigen (HLA) bound peptides presented specifically on the surface of malignant cells[2]. Despite unprecedented developments exploring the tumor immunopeptidome repertoire, most studies remain focused on canonical antigens, derived from protein-coding genomic regions, such as mutated neoantigens which are mostly patient-specific[1,3,4]. Immunogenic tumor-specific antigens that are shared across patients might provide a more promising approach in terms of treatment effectiveness, notably because several patients can benefit from the same immunotherapy treatment[5]. In recent years, mass spectrometry

[1]Ludwig Institute for Cancer Research, University of Lausanne, Lausanne, Switzerland. [2]Department of Oncology, Centre Hospitalier Universitaire Vaudois, Lausanne, Switzerland. [3]Agora Cancer Research Centre, Lausanne, Switzerland. [4]SIB Swiss Institute of Bioinformatics, University of Lausanne, Lausanne, Switzerland. [5]Department of Pathology, University of Michigan, Ann Arbor, MI, USA. [6]Center of Experimental Therapeutics, Department of Oncology, Centre Hospitalier Universitaire Vaudois, Lausanne, Switzerland. [7]Department of Computational Medicine and Bioinformatics, University of Michigan, Ann Arbor, MI, USA. ✉e-mail: michal.bassani@chuv.ch

(MS) based immunopeptidomics coupled with novel proteogenomic approaches identified novel canonical and non-canonical cancer-specific antigens resulting from genetic and epigenetic alterations during cancer progression[6,7], affecting the cellular transcriptome[8], translatome[9], proteome[10,11] and the antigen presentation machinery[12]. Remarkably, circRNA translation has also been proposed as an unexplored source of antigens in cancer[13].

circRNAs, initially thought to be a by-product of transcription, are covalently closed sequences of RNA. In humans, such non-polyadenylated transcripts are produced by a non-canonical splicing process, known as back-splicing, between two non-sequential exons where the 3′ end of a downstream exon is fused to the 5′ end of an upstream exon[14]. The generated junction is called back-splicing junction (BSJ). The first described circRNAs were composed only of exonic sequences, but the portfolio was expanded to include intronic and exonic-intronic circRNAs, containing introns and exons/introns, respectively. In general, the expression of most circRNAs is low compared to their linear counterparts, but some circRNAs are highly abundant and represent the main transcribed products of the host genes[15]. Because of their covalently closed-loop structures, circRNAs are protected from exonucleases. CircRNAs' inherent stability makes them highly promising biomarkers across diverse human diseases. For example, altered levels of circRNAs can be detected in urine and blood of cancer patients[16–19]. Their biogenesis is regulated by different RNA-binding proteins (RBPs), such as quaking (QKI) and adenosine deaminases acting on RNA (ADARs) proteins, which have been implicated in tumor progression[20–23].

Functionally, one of the most studied roles of circRNAs is their potential to act as a sponge for microRNAs (miRNAs), sequestering these regulatory molecules so that they no longer target mRNAs for degradation or translation inhibition[24]. However, circRNAs have additional gene expression regulatory functions. They compete with canonical splicing[25], enhance parental gene expression by interacting with the Pol II machinery[26,27] or act epigenetically by regulating DNA methylation and active DNA demethylation[28]. CircRNAs can also interact and sequester some RBPs, modulating their activity[29]. Despite being considered "non-coding" transcripts, lacking a 5′ cap and 3′ poly(A) tail, their sequences often harbor regulatory sequences which can promote cap-independent translation, driven by internal ribosome entry sites (IRES)[30] or consensus N⁶-methyladenosine-modified motifs[31,32]. Extensive analyses of the translation potential of endogenous circRNAs using MS data demonstrated the enrichment of IRES-like short elements in endogenous circRNAs able to initiate their translation[33]. Efficient circRNA translation has also been demonstrated in studies using exogenous circRNAs with infinite open reading frames (ORFs) lacking stop codons, undergoing rolling circle translation[34]. Accumulating evidence has shown the potential of circRNAs to be translated into functional proteins in cancer[35]. For example, the translation of a *CTNNB1* circRNA (*hsa_circ_0004194*) gives rise to a novel isoform of β-catenin, harboring a distinct, shorter C-terminus via the creation of a new stop codon after circularization. This isoform was implicated in tumor growth by activating the Wnt/β-catenin pathway[36], emphasizing the role of *CTNNB1* in cancer through non-genetic alterations by the creation and translation of a circRNA, rather than via activating mutations[37].

Importantly, circRNAs may encode peptide sequences spanning the BSJ that are distinct from those sequences generated by the canonical splicing process. However, as circRNAs are mostly predicted to have short ORFs, especially those spanning the BSJ, their detectability by MS-based proteomics and by ribosome profiling is challenging[38–41]. The lack of experimental evidence for these short ORFs at the proteome level could be explained by their possible higher instability[42] resulting in proteasomal degradation[33]. However, unstable proteins with short half-lives are ideal sources for HLA class I (HLA-I) peptides, and therefore, circRNAs could be an interesting source of neoantigens that might play a role in tumor immunosurveillance[42]. A combination of ribo-seq profiling and shotgun proteomics MS have been used to predict the circRNA-derived translatome and associated proteome[39,43]. Furthermore, in a recent study, the discovery of putative circRNA-encoded proteins was accompanied by the detection of two HLA-I-associated peptides[44], but neither of them was encoded by the region overlapping the BSJ, therefore, they could potentially derive from the linear transcript. In another study, 13 predicted circRNA-derived antigens from hepatobiliary tumor organoids were validated with MS. However, interpretation of the results is difficult since an unusual sample processing method was used, in which, following HLA-I complex purification, the eluates were submitted to gel electrophoresis (SDS-PAGE) and in-gel trypsin digestion prior to peptide cleanup and MS analysis[45], steps that are expected to be deleterious for HLA bound peptides and are not typically employed in immunopeptidomics[46].

In this study we developed an approach to identify HLA-presented circRNA-derived peptides by MS immunopeptidomics. We focused on melanoma and lung cancer as tumor models where aberrant expression of circRNAs has been already documented[47,48]. The workflow included the design and generation of a generic reference database containing trimmed circRNA-derived ORFs spanning the BSJ and initiated by the canonical start codon ATG, allowing the MS-based identification of circRNA-derived antigens overlapping the BSJ. Validation of the MS-detected peptides was performed by parallel reaction monitoring (PRM) and the presence of the corresponding source circRNAs transcripts was confirmed by divergent RT-PCR and Sanger sequencing. We identified 29 different circRNA-derived peptide sequences spanning the BSJ in two melanoma samples. After treatment with IFNγ or the proteasome inhibitor MG132, the presentation of circRNA-derived peptides was controlled in a manner comparable to canonical peptides. Furthermore, we discovered 21 unique circRNA-derived HLA-I and HLA-II peptide sequences spanning the BSJ in a cohort of eight lung cancer patient tumors. We discussed challenges associated with the detection of circRNA-derived immunopeptides and the importance of exploring their presentation across tumor and healthy tissues. Our approach enabled the identification of tumor-associated antigens encoded by circRNAs, that have the potential to be promising targets for immunotherapy.

## Results

### A dedicated workflow for the detection of circRNA-derived HLA peptide candidates

To explore the presentation of unique peptides potentially derived from circRNA sources spanning the BSJ that are not found in any of the translation frames of the canonical linear gene transcripts, we generated a reference fasta file of circRNA sequences present in circBase[49] (Fig. 1a). circBase is the first repository of circRNAs which merged different datasets of circRNAs, offering an interface with standardized annotations and unique identifiers. Using a targeted approach, we identified and in silico translated all BSJ-containing "stop-to-stop" circRNA fragments that had a canonical translation initiation codon ATG upstream of the BSJ (see Methods section and Supplementary Fig. 1). Stop-to-stop sequences that did not contain this codon were discarded. HLA-I peptides are short (8-15 amino acids, mostly 9 mers), while HLA-II peptides may reach up to 25 amino acids (average length of around 15-16 amino acids). Therefore, where possible, sequences were further trimmed to a length of with up to 49 amino acids covering the transcript position corresponding to at least one BSJ (24 amino acids upstream the BSJ, one amino acid partially encoded by the BSJ and 24 amino acids downstream the BSJ). This made the circRNA-derived BSJ-ORF fasta reference suitable for both HLA-I and HLA-II MS-based immunopeptidomics workflows (Fig. 1b; see Methods section)[50]. The trimmed circRNA-derived putative ORF fasta sequences spanning the BSJ and initiated by the canonical start codon ATG were

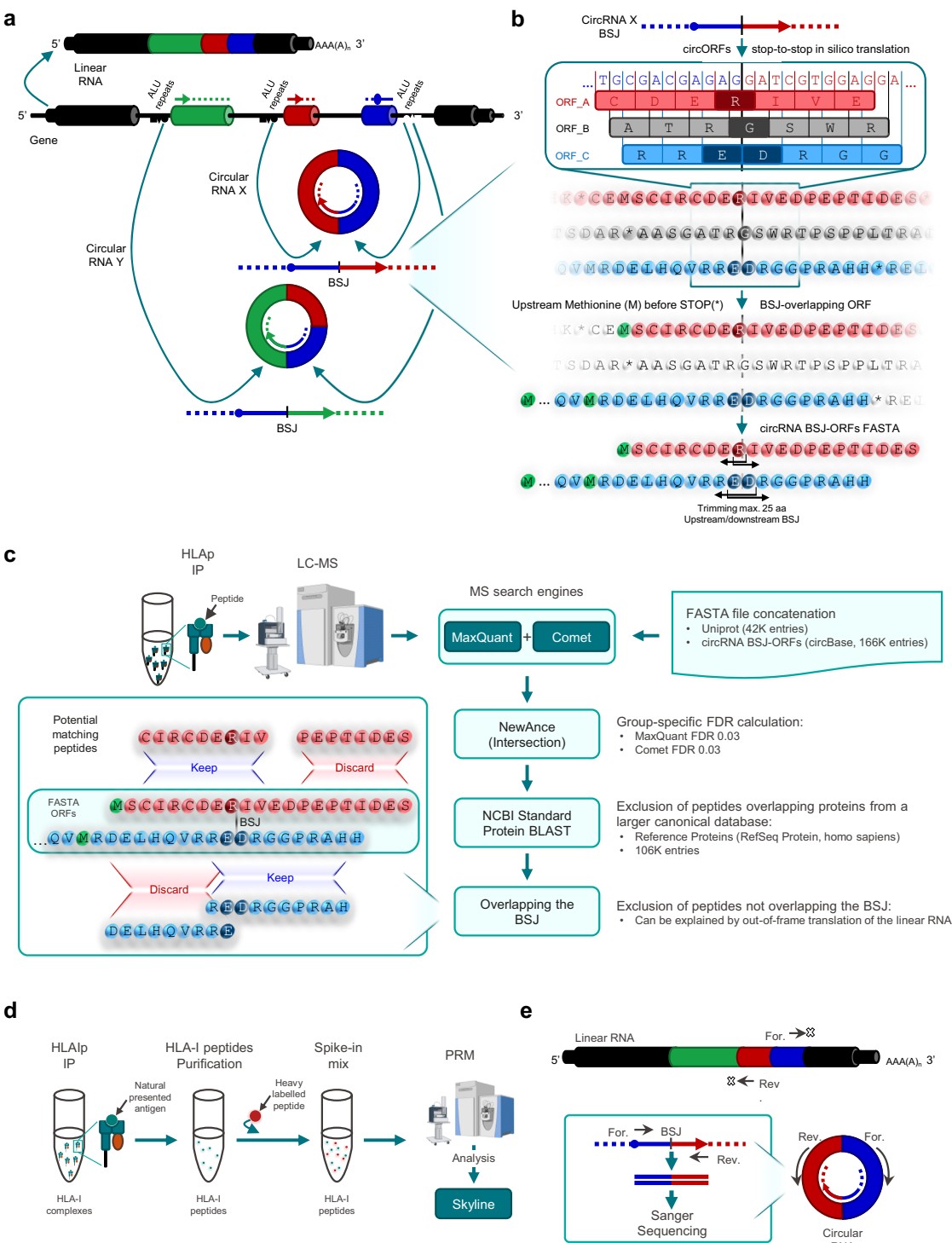

concatenated with a human UniProt fasta file[51], containing canonical protein sequences, before performing the MS database search. To apply stringent cutoffs and to minimize false identifications, the immunopeptidomics MS raw files were initially searched against the concatenated fasta reference with two search engines, MaxQuant[52] and Comet[53]. The NewAnce tool[6] was used to calculate a group-specific FDR of 0.03, for both MaxQuant and Comet, for peptides derived from the human UniProt entries (namely group protein-coding 'PC') and peptides derived uniquely from the circRNA-derived BSJ-ORF fasta reference (group 'circRNA'). Only peptides identified by both search engines were retained. In this study, we were particularly interested in

characterizing circRNA-derived peptides that span the BSJ, named 'circRNA-BSJ'. Importantly, our searches against the circRNA group resulted in identification of other circRNA-derived peptides that do not overlap the BSJ (specifically named below as 'circRNA-not-BSJ' when applicable), that could potentially derive from out-of-frame translation events of the matched linear RNAs (with canonical or alternative translational initiation sites), which also represent a potential source of non-canonical antigens. Their inclusion in the study was not within the study's intended scope. They were added for the sake of comparison and completeness. Furthermore, an additional step of peptide mapping against the NCBI Standard Protein BLAST

**Fig. 1 | Immunopeptidomics workflow for identification and validation of circRNA-derived HLA-I bound peptides. a** Canonical splicing generates a linear mRNA molecule comprising a 5' cap and 3' poly-A tail which assist the canonical initiation of translation. Through back-splicing, a downstream 5' splice site is joined to an upstream 3' splice site, producing a covalently closed structure called circRNA. In exonic circRNAs, back-splicing is associated with the presence of complementary ALU repeats in their flanking long introns[15]. One host gene can generate multiple circRNAs. circRNAs can undergo cap-independent translation. For illustration purposes, a circRNA composed exclusively of exons is shown and the different back-spliced exons are colored. **b** Fasta file construction for MS search for the identification of circRNA-derived peptides. Sequences flanking the BSJ of circRNAs were extracted from circBase and in silico translated in three forward frames. Only BSJ overlapping translated ORFs were kept, having a methionine (canonical initiation translation codon ATG) encoded upstream of the BSJ (colored in green). ORFs sequences were trimmed to a maximum of 25 amino acids

upstream and downstream of the amino acid(s) encoded within the BSJ. Workflow is illustrated using a fictitious transcript sequence derived from an exonic circRNA, but the strategy was applied to all human circRNAs in circBase, regardless of their annotation. The different frames of translation are differently colored and the amino acids spanning the BSJ are highlighted with a darker color. **c** After immunoaffinity purification of HLA complexes, peptides were purified and analyzed by LC-MS/MS. Peptide identification was performed by two search engines against the above mentioned circRNA-derived ORFs database concatenated with a Human Uniprot database and applying a group-specific FDR. NCBI Standard Protein Blast was used to exclude peptides matching known proteins found in the RefSeq human protein database. Only peptides overlapping the BSJ were kept. **d** PRM validation was used to confirm the presence of the candidate peptides with heavy peptides spiked in the matrix. **e** At transcript level, the expression of the circRNAs was confirmed by divergent RT-PCR followed by direct Sanger Sequencing. Some elements from panels c and d were created with BioRender.com.

database was performed to remove peptides mapping to annotated coding sequences within this much larger reference (Fig. 1c). Experimental validations of peptide identification were carried out by introducing heavy-labeled synthetic peptides into newly generated immunopeptidome samples, followed by PRM analyses (Fig. 1d). Furthermore, the confirmation of the presence of circRNA transcripts generating the back-splicing event was accomplished through divergent RT-PCR, with subsequent direct Sanger sequencing of the amplicons (Fig. 1e).

HLA-I peptides eluted from the T1185B melanoma cell line were analyzed by LC-MS/MS with data-dependent acquisition (DDA). In total, we identified 17,770 PC peptides, 122 circRNA peptides, including 19 circRNA-BSJ peptides that overlap the BSJ by one or two amino acids, depending on the relative position between the codons and the BSJ (Table 1, Fig. 2a). PC derived peptides exhibited the expected length distribution for HLA-I peptides (average length of 9.80), while circRNA, and the subset of circRNA-BSJ peptides were overall longer (average length of 10.07 and 10.11, respectively; Fig. 2b). Indeed, extended HLA-I restricted peptides, longer than 11 amino acids, effectively stimulate CD8$^+$ T cell responses within the typical range for epitope-specific CD8 + T cells, however, longer, "bulging" peptides may pose challenges for T-cell receptor recognition compared to shorter peptides[54]. 91%, 95% and 89% of the PC, circRNA, and circRNA-BSJ derived peptides, respectively, were predicted to bind any of the HLA-I molecules expressed in T1185B cells with a rank threshold < 2% (Fig. 2c). Overall, circRNA-BSJ spanned the entire range of peptide intensities (Fig. 2d), with 32% of circRNA-BSJ peptides consistently detected in all three biological replicates (Fig. 2e). This percentage is within the range observed for PC (36%) and all peptides derived from circRNA (32%) groups. We observed a global enrichment in HLA-A*68:01 bound PC peptides that was even more profound in the circRNA and circRNA-BSJ groups (Fig. 2f).

Comparable results were obtained when the workflow was applied to another cell line and matched tumor tissue derived from melanoma patient Mel-1. Here, 9,666 PC peptides and 61 circRNA-derived peptides were detected in the Mel-1 cell line, while in the tumor tissues, 15,443 PC and 55 circRNA-derived peptides, respectively, were identified (Supplementary Fig. 2a). In total, 11 unique circRNA-BSJ peptides were identified in the Mel-1 samples (Table 1). The expected peptide length distribution and high percentage of HLA-binders for both PC and circRNA groups confirmed the high quality of the data (Supplementary Fig. 2b, c), and the intensity of circRNA-BSJ peptides was found within the range of all other PC peptides (Supplementary Fig. 2d). With one exception of a peptide predicted to bind HLA-A*02:01 molecule (identified in both cell line and tumor tissue), all other circRNA-BSJ peptides were predicted to bind HLA-A*03:01. The HLA allotype distribution of the peptides predicted to bind the respective HLA molecules was remarkably different between the Mel-1 cell line and the matched tumor tissue (Supplementary Fig. 2e),

suggesting a dysregulation of the HLA expression in the expanded primary cell line. Nevertheless, a clear enrichment in HLA-A*03:01 bound circRNA-derived peptides was observed in both sample types.

## Validation approaches for candidate circRNA-BSJ derived peptides

The resulting spectrum matches of candidate circRNA-BSJ peptides were manually checked using PDV spectrum visualization tool[55] (Supplementary Fig. 3). Furthermore, to validate the identification of peptides, we spiked heavy-labeled synthetic peptides corresponding to our candidates into newly generated immunopeptidome samples from T1185B and Mel-1 cell lines. We subjected the cell line samples to PRM analyses and compared the co-elution profile and the fragmentation pattern of the synthetic heavy-labelled and endogenous light peptides. Applying this method, we validated seven and four unique circRNA-BSJ derived peptides in T1185B and Mel-1 cell lines, respectively (Table 1, Supplementary Fig. 4). Furthermore, the presence of circRNA transcripts generating the back-splicing event in the samples was confirmed with a divergent RT-PCR followed by direct Sanger sequencing of the amplicons. In total, the back-splice junctions of eleven and three circRNAs were validated in T1185B and Mel-1, respectively (Table 1, Supplementary Figs. 5, 6).

As an example, the circRNA-BSJ peptide DLYNGSSIVS[R], predicted to bind HLA-A*68:01 was identified in the T1185B cell line. The square brackets in the peptide sequence above represent the amino acid encoded by the codon spanning the BSJ. By PRM we detected the co-elution of the heavy labelled peptide and the endogenous light peptide and the similar MS/MS fragmentation patterns (Fig. 2g and Supplementary Fig. 4). This peptide matched four potential circRNA-derived ORFs: hsa_circ_0015364, hsa_circ_0015366, hsa_circ_0111261 and hsa_circ_0111262, all of them hosted by gene *COP1* (also known as *RFWD2*) (Fig. 2h). Divergent RT-PCR followed by direct Sanger sequencing confirmed that at least two of those circRNAs are expressed in T1185B (Fig. 2i, j), representing two possible sources for this HLA peptide.

The circRNA-BSJ peptide [R]VFEVYHTTVLK, matching the circRNAs *hsa_circ_0003137* and *hsa_circ_0004030* from *CTNNB1* gene, was detected and validated with PRM and Sanger sequencing in both T1185B and Mel-1 samples (Supplementary Fig. 4, Supplementary Fig 5, and Supplementary Fig. 6), and its shorter circRNA-not-BSJ version EVYHTTVLK was also detected. An additional shorter peptide VYHTTVLK and the partially overlapping peptide HTTVLKIQR from the same potential circRNA ORFs, were detected in Mel-1 and T1185B cell lines, respectively. Interestingly, [R]VFEVYHTTVLK and EVYHTTVLK were previously reported to be detected in melanoma as well as benign human tissues and to be derived potentially from translation of a novel upstream ORF in *CTNNB1* gene through an alternative translational initiation site[56]. Our results suggest a potential circRNA source, that importantly, contains the canonical translation initiation ATG codon.

**Table 1 | HLA-I circRNA-derived peptides, overlapping the BSJ encoding region, detected in T1185B and Mel-1**

| Sample | Gene ID | circRNAs_frame | Infinite translation candidate | Peptide Sequence | Length (AA) | NetMHCpan – 4.1 Binder/ Best Allele | BSJ context (3'Exon -> Peptide [BSJ] <- 5'Exon) | PRM validation | Divergent PCR/ Sanger Sequencing | Detection |
|---|---|---|---|---|---|---|---|---|---|---|
| T1185B | ANXA6 | hsa_circ_0074614_0 | TRUE | VQDLIADLK | 9 | no_binder | VVLLQGTREEDDVVSEDLVQQD-> V[QD]LIADLK <- -YELTGKFERLIVGLMRP | – | – | CL |
| | CBLL1 | hsa_circ_0081903_0 | FALSE | DTFFGTFR | 8 | HLA-A*68:01 | GMTVKGVSCLQISEDFL-> DTFFGTF[R] <- -RI | – | – | CL |
| | CDC73 | hsa_circ_0111569_0 | TRUE | TTENIPVVRR | 10 | HLA-A*68:01 | KEETEGFKIDTMGTYHGMTLKSV-> [TT]ENIPV-VRR <- -PDRKDLLGYLNGEAS | + | + | CL |
| | CLASP2 | hsa_circ_0123550_2 | FALSE | LETLGDKECI | 10 | no_binder | SFSVWDEHFKTILLLL-> LETLGDK[EC]I <- | NT | – | CL |
| | CTNNB1 | hsa_circ_0003137_1 | FALSE | RVFEVYHTTVLK | 12 | HLA-A*68:01 | MATKKA-> [R]VFEVYHTTVLK <- -IQRGQWLLKLI | + | + | CL |
| | | hsa_circ_0004030_1 | FALSE | | | | | SWMGCLQVTAISWPGLLTCKSSF-> [R]VFEVYHTTVLK <- -IQRGQWLLKLI | | + | |
| | CTSB | hsa_circ_0083357_1 | FALSE | STMSTNGIPRGR* | 12 | HLA-A*68:01 | GAGPLSIPCRMSW-> STMSTNGIPRG[R] <- -GRGREGT | Inconclusive | – | CL |
| | | | | STNGIPRGR | 9 | HLA-A*68:01 | GAGPLSIPCRMSWSTM-> STNGIPRG[R] <- -GRGREGT | + | – | CL |
| | ETV5 | hsa_circ_0068440_2 | TRUE | FVPDFQSDNR | 10 | HLA-A*68:01 | QEAWLAEAQVPDDEQ-> FVPDFQSDN[R] <- -GSLFPQKLLNAETSQSGIRDAEST | – | + | CL |
| | GAS7 | hsa_circ_0042080_0 | FALSE | EAWNGPPSAGR | 11 | HLA-A*68:01 | WERPSSSPGIPASPGSHRSSLPPT-> [E]AWNGPPSAGR <- -RKPDGHPSTWLAEL | + | – | CL |
| | KIAA0226 | hsa_circ_0068719_1 | FALSE | SSITGLSVTR | 10 | HLA-A*68:01 | MVAWSGFAGTCRA-> SSITGLSVT[R] <- -REHWQLLGNLKTTVEGLVSTNSPN | + | – | CL |
| | LIN9 | hsa_circ_0016709_0 | FALSE | EIYSNNVNTR | 10 | HLA-A*68:01 | ADVSQFKDLPDEIPLPLVIGTKVT-> [EI]YSNNVNTR <- | – | – | CL |
| | LINC00221 | hsa_circ_0101415_1 | FALSE | AAAAFRFPR | 9 | HLA-A*68:01 | MPARSWLGE-> AAAAFRFP[R] <- -DFCVCAHIDVPTLCPSQ | + | + | CL |
| | | | | EAAAAFRFPR | 10 | HLA-A*68:01 | MPARSWLG-> EAAAAFRFP[R] <- -DFCVCAHIDVPTLCPSQ | – | – | CL |
| | | hsa_circ_0101416_0 | FALSE | AAAAFRFPR | 9 | HLA-A*68:01 | MPARSWLGE-> AAAAFRFP[R] <- -M | + | + | CL |
| | | | | EAAAAFRFPR | 10 | HLA-A*68:01 | MPARSWLG-> EAAAAFRFP[R] <- -M | (+) | + | CL |
| | NOD1 | hsa_circ_0079703_1 | FALSE | VPAPPSLTR | 9 | HLA-A*68:01 | MTTSRPKMRRLC-> VPAPPSLT[R] <- -GCAE | NT | – | CL |
| | PAFAH1B2 | hsa_circ_0095083_1 | FALSE | STTDLFWTVK | 10 | HLA-A*68:01 | MENWRILSL-> [S]TTDLFWTVK <- -TKSLMYCSWETPWCS | Inconclusive | – | CL |
| | PKNOX1 | hsa_circ_0061852_2 | FALSE | QTPFAFHPR | 9 | HLA-A*68:01 | VVTPQGGVVTQTLSPGTIRIQNS-> [QT]PFAFHPR <- | Inconclusive | + | |
| | | hsa_circ_0116109_2 | FALSE | | | | | KTKMNSETLLSGEPGSPSYSPVQS-> [QT]PFAFHPR <- | | + | |
| | | hsa_circ_0116112_2 | FALSE | | | | | TPVNMNVDSLQSLSSDGATLAVQ-> [QT]PFAFHPR <- | | + | |
| | PTPRG | hsa_circ_0066402_0 | FALSE | NTASMAGGFLLR* | 12 | HLA-A*68:01 | MAQRAL-> NTASMAGGFLL[R] <- -HVTSCMEAWPVSSPALFVPSQAP | Inconclusive | – | CL |
| | | hsa_circ_0066406_0 | FALSE | | | | | MAQRAL-> NTASMAGGFLL[R] <- -CLWS | | + | |
| | | hsa_circ_0124401_0 | FALSE | | | | | MAQRAL-> NTASMAGGFLL[R] <- -YYRTCSD | | – | |
| | RFWD2(COP1) | hsa_circ_0015364_0 | FALSE | DLYNGSSIVSR | 11 | HLA-A*68:01 | YNSVRPLATLSYAS-> DLYNGSSIVS[R] <- -QRNSLGIIAR | + | – | CL |
| | | hsa_circ_0015366_0 | FALSE | | | | | YNSVRPLATLSYAS-> DLYNGSSIVS[R] <- -NHMQPNYRFLWNSSRLQEEIRESN | | + | |
| | | hsa_circ_0111261_0 | FALSE | | | | | YNSVRPLATLSYAS-> DLYNGSSIVS[R] <- -K | | – | |
| | | hsa_circ_0111262_1 | FALSE | | | | | YNSVRPLATLSYAS-> DLYNGSSIVS[R] <- -TSGTMMNRSGDSGAS | | + | |
| | STAT3 | hsa_circ_0043818_1 | TRUE | TIYEESSSFFR | 11 | HLA-A*68:01 | SIAASCKSRMFSIS-> TIYEESSSFF[R] <- -VSGAAAAQTGRGSRGFRRRSRGNK | Inconclusive | – | CL |
| | | hsa_circ_0106859_1 | FALSE | | | | | SIAASCKSRMFSIS-> TIYEESSSFF[R] <- -MAQWNQLQQLDTRYLEQLHQLYSD | | – | |
| Mel-1 | ABCA2 | hsa_circ_0089621_2 | FALSE | RLKEPSTQR | 9 | HLA-A*03:01 | ISWVYSVAMTIQHIVAEKEH-> RLK[EP]STQR <- -RP | NT | – | Tumor |
| | CCAR1 | hsa_circ_0018553_1 | FALSE | VLSKGKPPK | 9 | HLA-A*03:01 | ISAASITPLLQTQPQPLLQQPQQK-> [VL]SKGKPPK <- | NT | – | CL |
| | CEP57L1 | hsa_circ_0130265_1 | FALSE | KLRDPTDSTLR | 11 | HLA-A*03:01 | MGLSLSCKN-> KLRDPTDSTL[R] <- -AWYLVQT | NT | – | CL |
| | CTNNB1 | hsa_circ_0003137_1 | FALSE | RVFEVYHTTVLK | 12 | HLA-A*03:01 | MATKKA-> [R]VFEVYHTTVLK <- -IQRGQWLLKLI | + | + | CL + Tumor |
| | | hsa_circ_0004030_1 | FALSE | | | | | SWMGCLQVTAISWPGLLTCKSSF-> [R]VFEVYHTTVLK <- -IQRGQWLLKLI | | + | |
| | MALAT1 | hsa_circ_0096124_2 | FALSE | KLLHGVKNVFK | 11 | HLA-A*03:01 | M-> [KL]LHGVKNVFK <- -RKLRERTTEPRINT | Failed QC | – | CL + Tumor |
| | PDIA3 | hsa_circ_0035034_2 | FALSE | RVSSYFVLH | 9 | HLA-A*03:01 | LLERFLLLLSELLKERSLSCRRSS-> [R]VSSYFVLH <- -ISLTSLRTRLWHIQSK | NT | – | CL |
| | SAE1 | hsa_circ_0051662_2 | FALSE | ILAQEIVKV | 9 | HLA-A*02:01 | MAPVCAVVGG-> ILAQEIV[KV] <- -ILLRDGPSVCGWRDFGTGNCEG | NT | – | CL + Tumor |
| | PPP2R3A | hsa_circ_0122039_2 | FALSE | KVSLSFTEK | 9 | HLA-A*03:01 | SPVGDKAKDTTSAVLIQQTPEVI-> [KV]SLSFTEK <- | + | – | CL + Tumor |
| | TOPBP1 | hsa_circ_0121989_1 | FALSE | KSLAAELLVLK | 11 | HLA-A*03:01 | MYTPHCART-> K[S]LAAELLVLK <- | + | + | CL + Tumor |
| | WARS | hsa_circ_0033185_2 | FALSE | VLTRLLNK | 8 | HLA-A*03:01 | MPRTSSPVALTSTRLSYSLTWTTW-> [V]LTRLLNK <- -SSDLRPAVNVVPER | – | – | Tumor |
| | | hsa_circ_0033189_1 | FALSE | | | | | SDGESAALESARPAPLLILPVCM-> [V]LTRLLNK <- -SSDLRPAVNVVPER | | – | |
| | ZMYM4 | hsa_circ_0113154_2 | FALSE | RVVVSWIQK | 9 | HLA-A*03:01 | DKAANQVEETLHTHLPQTPETNF-> [RV]VVSWIQK <- -CLKI | + | – | Tumor |

Amino acids spanning the BSJ are represented within squared brackets and bold font. PRM validation of peptide duplicates from an alternative circRNA source are shown within parenthesis. In T1185B, seven different peptides were validated by PRM, four of which could be encoded by at least one circRNA validated by Sanger sequencing. In Mel-1 cell line, four different peptides were validated by PRM, two of which could be encoded by at least one circRNA validated by Sanger sequencing. (*): Peptide also detected with a Methionine Oxidation; "+": validated; "–": not validated, NT Not tested, CL Cell line.

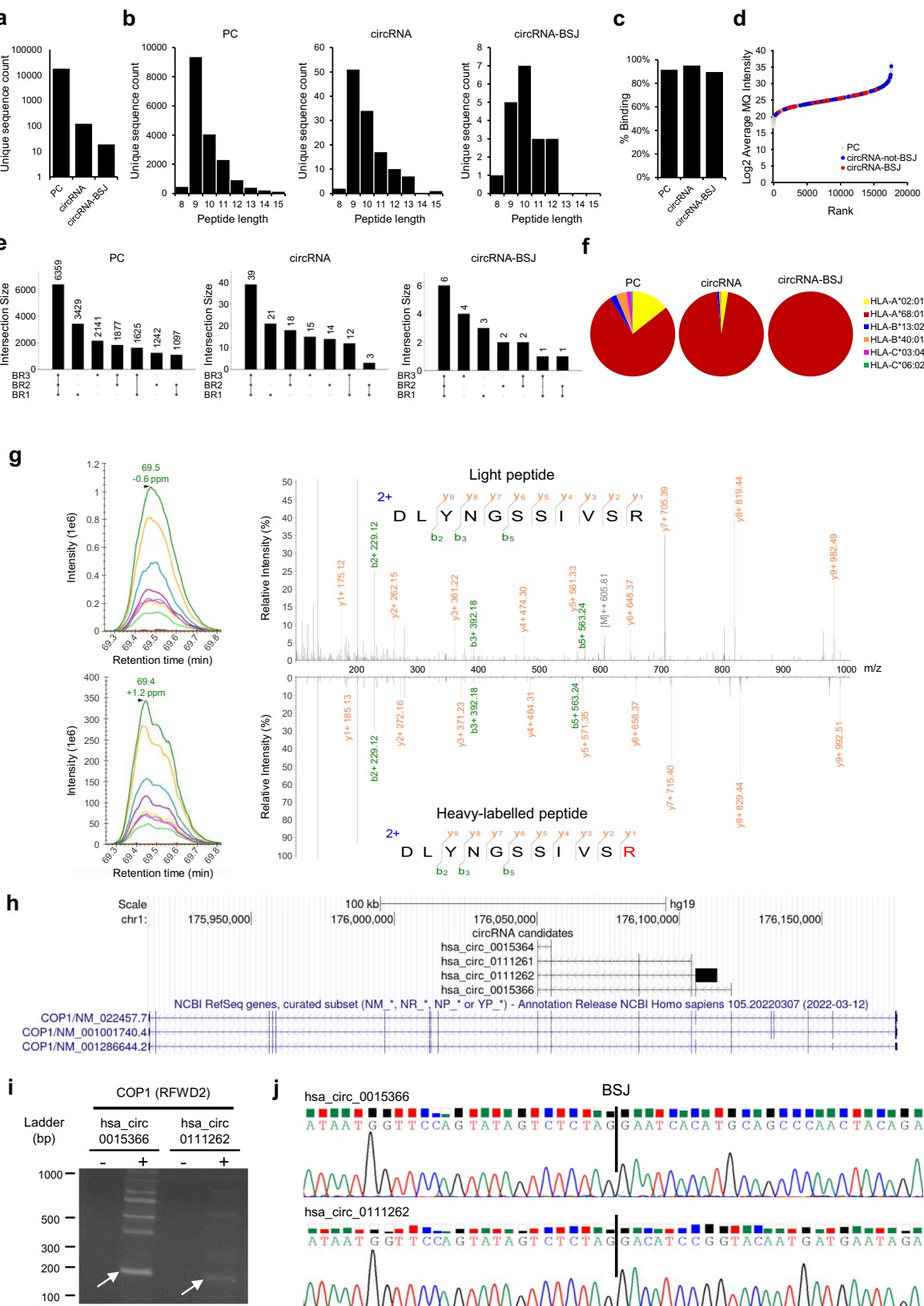

Notably, although not directly related, an amplicon obtained through unspecific amplification during RT-PCR of the *CTNNB1 hsa_circ_0003137*, matched the *CTNNB1* circRNA *hsa_circ_0004194* (Supplementary Figs. 5, 6) which doesn't encode for [R]VFE-VYHTTVLK. *hsa_circ_0004194* was reported to be associated with the translation of a novel isoform of β-catenin with a shorter C-terminus leading to the promotion of cell growth in hepatocellular carcinoma[36], suggesting that additional potential peptides may be derived from various novel coding ORFs in the *CTNNB1* gene.

**Fig. 2 | Validation of circRNA-derived peptides overlapping the back-splice junction. a** Total number of peptide sequences, (**b**) Peptide length distribution and (**c**) percentage of peptides predicted as HLA binders (% rank < 2), in T1185B, annotated in the protein (PC) and circRNA groups, and circRNA-BSJ subgroup. **d** Rank plots of the PC, circRNA-not-BSJ, circRNA-BSJ derived peptides intensities in T1185B cell line. **e** UpSet plots showing the overlap of detected peptides identified in PC, and circRNA groups, and circRNA-BSJ subgroup among the three biological replicates of T1185B cell line. **f** HLA restriction of PC, circRNA (groups), circRNA-BSJ (subgroup) derived binder peptides in T1185B cell line. **g** PRM validation of peptide DLYNGSSIVS[R] encoded by circRNAs hosted by *COP1* (*RFWD2*) gene. Skyline (left) and pLabel visualizations (right) showed the co-elution and similar MS/MS fragmentation patterns of endogenous light (above) and synthetic heavy-labelled

(below) peptides. To improve readability, MS/MS fragmentation pattern figures were edited and displayed as a mirror plot. **h** UCSC genome browser (hg19) visualization of the four circRNAs hosted by *COP1* (*RFWD2*) which can potentially encode peptide DLYNGSSIVS[R] within an ORF overlapping their respective BSJs. **i** Agarose gel electrophoresis of divergent RT-PCR products validating the expected amplicons (size) in two of the *COP1* (*RFWD2*) circRNAs, *hsa_circ_0015366* and *hsa_circ_0111262*. Arrows indicate the amplicons that were selected for direct Sanger Sequencing. The image was adjusted to improve legibility. This experiment was performed once. **j** Sanger sequencing demonstrating the specific nucleotide sequence upstream and downstream the BSJ for the above circRNAs. Source data are provided as a Source Data file.

Another interesting example is the *hsa_circ_0111569* circRNA generated by back-splicing between exon 10 - exon 3 of the *CDC73* (*NM_024529.5*) host gene (Fig. 3a). This circRNA has an ORF of 735 nucleotides with multiple start codons but without any stop codon (Fig. 3b), having the potential for rolling circle translation[34]. This unique ORF encodes for the circRNA-BSJ peptide [TT]ENIPVVRR, a predicted HLA-A*68:01 ligand, that we identified and validated by PRM in the T1185B immunopeptidome (Fig. 3c and Supplementary Fig. 4). Furthermore, the presence of this junction was validated through RT-PCR and Sanger sequencing in T1185B samples (Fig. 3d, e). Curiously, in the canonical protein, the two exon junctions involved in the back-splicing event are also spanned by two other MS-identified canonical PC peptides (AT)ENIPVVRR and SV(TE)GASAR, where the amino acids encoded by the two codons spanning the linear spliced junctions are represented above within the brackets. These two peptides are also predicted to bind HLA-A*68:01, and we validated them with PRM (Fig. 3c and Supplementary Fig. 4). Both peptides overlap the linear exon-exon junctions, suggesting that this translated region may have some features that promote its processing and presentation.

## Comparable presentation levels of circRNA-BSJ and PC peptides post IFNγ treatment or proteasome inhibition

To further explore the biogenesis of circRNA-BSJ derived HLA-I bound peptides, we treated T1185B cells with either IFNγ or the proteasome inhibitor MG132. A data-independent acquisition (DIA) MS acquisition workflow was applied to compare the presentation levels of peptides derived from circRNA and PC sources (Fig. 4a). To properly control false identifications and to properly account for lower likelihood of true identification of non-canonical sequences[57], we implemented a group-specific FDR control in FragPipe[58–61]. When calculating the FDR, canonical and non-canonical peptides were classified into different groups. The FDR was calculated for each group separately because different groups have different score distributions. Here, we applied a group-specific FDR of 0.03, allowing stringent control of error in the circRNA search space. Furthermore, to increase stringency of our analysis, hybrid DIA analyses were performed also using Spectronaut with global FDR of 0.01, and only peptides identified by both tools were retained for further analysis (Fig. 4b and Supplementary Fig. 7a–d). Using this approach, 25,284 PC and 27 unique circRNA-BSJ peptides were identified (Supplementary Data 1), 16 of them were in common with the DDA analysis described above. Again, the HLA allelic distribution revealed a prominent presentation of HLA-A*68:01 bound peptides in the PC and circRNA peptidomes (Fig. 4b, c). Intensity values of all quantified peptides clustered hierarchically based on the treatment (see Methods section, Supplementary Fig. 7e), therefore, we performed a differential presentation analysis. We detected a highly significant increase in the presentation of the HLA-B*40:01 bound peptides following IFNγ treatment (Fig. 4d and Supplementary Fig. 8), in agreement with previous studies that demonstrated preferential IFNγ-induced upregulation of HLA-B expression[62]. Additionally, gene ontology enrichment analysis comparing the source protein annotation of the enriched peptides with the global distribution (using the

Student's t-test difference between treatment groups) revealed a significantly higher presentation of peptides derived from proteins associated with response to type I interferon following IFNγ treatment (presentation enrichment score: 0.44, Benj. Hoch. FDR: 1.15E-8; Fig. 4e and Supplementary Data 2). The proteasome inhibitor MG132 treatment led to an enrichment of peptides mapping essentially to proteasome subunits (KEGG term: Proteasome, presentation enrichment score: 0.32, Benj. Hoch. FDR: 8.52E-06; Fig. 4f and Supplementary Data 3) and had a broad impact on the peptidome, by increasing and decreasing the presentation of a large fraction of HLA-A*68:01 bound peptides (Supplementary Fig. 8). We found a minor, yet significant, downregulation of circRNA-derived peptides compared to PC peptides, upon IFNγ treatment (One-way ANOVA and Sidak's multiple comparisons test, adjusted *p*-value = 0.0178; Fig. 4g), mainly related to the relative increased presentation of HLA-B, and to a lower extent also of HLA-C alleles, that do not mediate presentation of circRNA-BSJ peptides that are associated with HLA-A*68:01. Indeed, by subsequently restricting the analysis only to the ligands of HLA-A*68:01, we found no significant difference in the presentation of circRNA-BSJ peptides following IFNγ or MG132 treatments (One-way ANOVA and Sidak's multiple comparisons test, adjusted *p*-value = 0.9976 for IFNγ and adjusted *p*-value = 0.4064 for MG132 treatments; Fig. 4g, h). Overall, our analysis suggests that circRNA-BSJ and PC peptides are similarly sampled for processing and presentation.

## Identification of potentially unique lung cancer associated circRNA-BSJ derived peptides

CircRNAs-derived peptides, especially those that span the BSJ, are an interesting potential source of neoantigens that can be used to precisely target cancer cells. However, defining their tissue specificity poses a challenge. Therefore, we leveraged recently published large HLA-I and HLA-II immunopeptidomics DIA and DDA datasets of tumoral and adjacent healthy matched multi-region tissues from eight lung cancer patients[63] and searched for circRNA-BSJ derived peptides presented specifically, or to a higher extent, in the tumors. With FragPipe, applying a group-specific FDR of 0.03, we built spectral libraries with the DDA data and searched the DIA data against it (Fig. 5a). Concerning HLA-I immunopeptidomes, overall, 119,084 peptide sequences were identified, ranging from 21,887 to 43,566 peptides across the patient's tumors and 16,637–34,888 peptides in the adjacent healthy tissues. Correspondingly, 42–154 and 32–128 circRNA-derived peptide sequences were identified in the tumor and healthy tissues, of which 19 were circRNA-BSJ (Fig. 5b, c, Supplementary Fig. 9, and Table 2). Fifteen of these circRNA-BSJ peptides were predicted to bind with affinity rank < 2% any of the HLA-I molecules expressed in at least one of the patients (Supplementary Data 4). Although none of these were deemed cancer-related based on their host gene expression in TCGA/GTEx (see Methods), six circRNA-BSJ peptides were uniquely detected in the tumor tissues, including the peptide ILDKKVE[KL]. This peptide is predicted to be encoded by the circRNA *hsa_circ_0076651* which is another example of an ORF with putative infinite translation, hosted by the *HSP90AB1* gene.

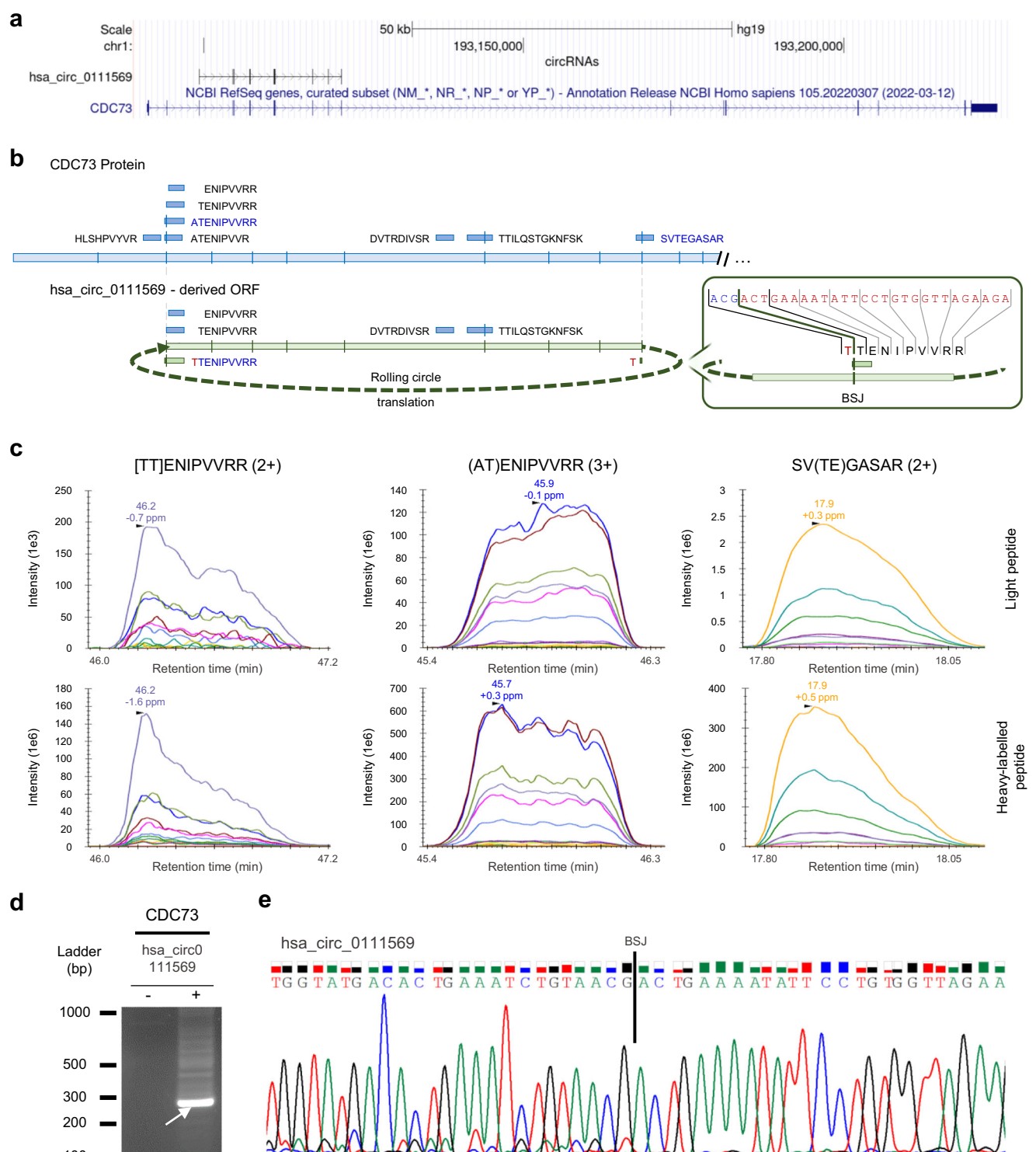

**Fig. 3 | Antigen presentation of *CDC73* gene. a** UCSC genome browser (hg19) visualization of *hsa_circ_0111569*, hosted by *CDC73* gene, which is predicted to have rolling circle translation. **b** Location of the nine peptides derived from gene *CDC73* that were detected by MS in T1185B cell line. *hsa_circ_0111569* derived peptide [TT] ENIPVVRR (green box) spans the BSJ corresponding to the exon 10 – exon 3 of *CDC73* gene. The remaining peptides (blue boxes) were annotated as canonical CDC73-derived peptides. Four of these peptides can be potentially derived from both CDC73 protein and the putative infinite circRNA ORF. In red, representation of the amino acid encoded upstream of the BSJ. In blue, selection of the peptides that underwent PRM validation. **c** PRM visualization with Skyline of the co-elution and similar MS/MS fragmentation patterns of light (above) and heavy-labelled (below) peptides after PRM. Validation was performed for the circRNA-derived peptide [TT] ENIPVVRR and the canonical (AT)ENIPVVRR and SV(TE)GASAR peptides. Square brackets indicate the BSJ and brackets the linear junction spanning amino acids. **d** Divergent RT-PCR validation. Agarose gel electrophoresis showed the expected amplicon (size) in *CDC73* circRNA. Arrow indicates the amplicon that was selected for direct Sanger sequencing. The image was adjusted to improve legibility. This experiment was performed once. **e** Sanger sequencing showed the specific sequence upstream and downstream the BSJ for the *hsa_circ_0111569*. Source data are provided as a Source Data file.

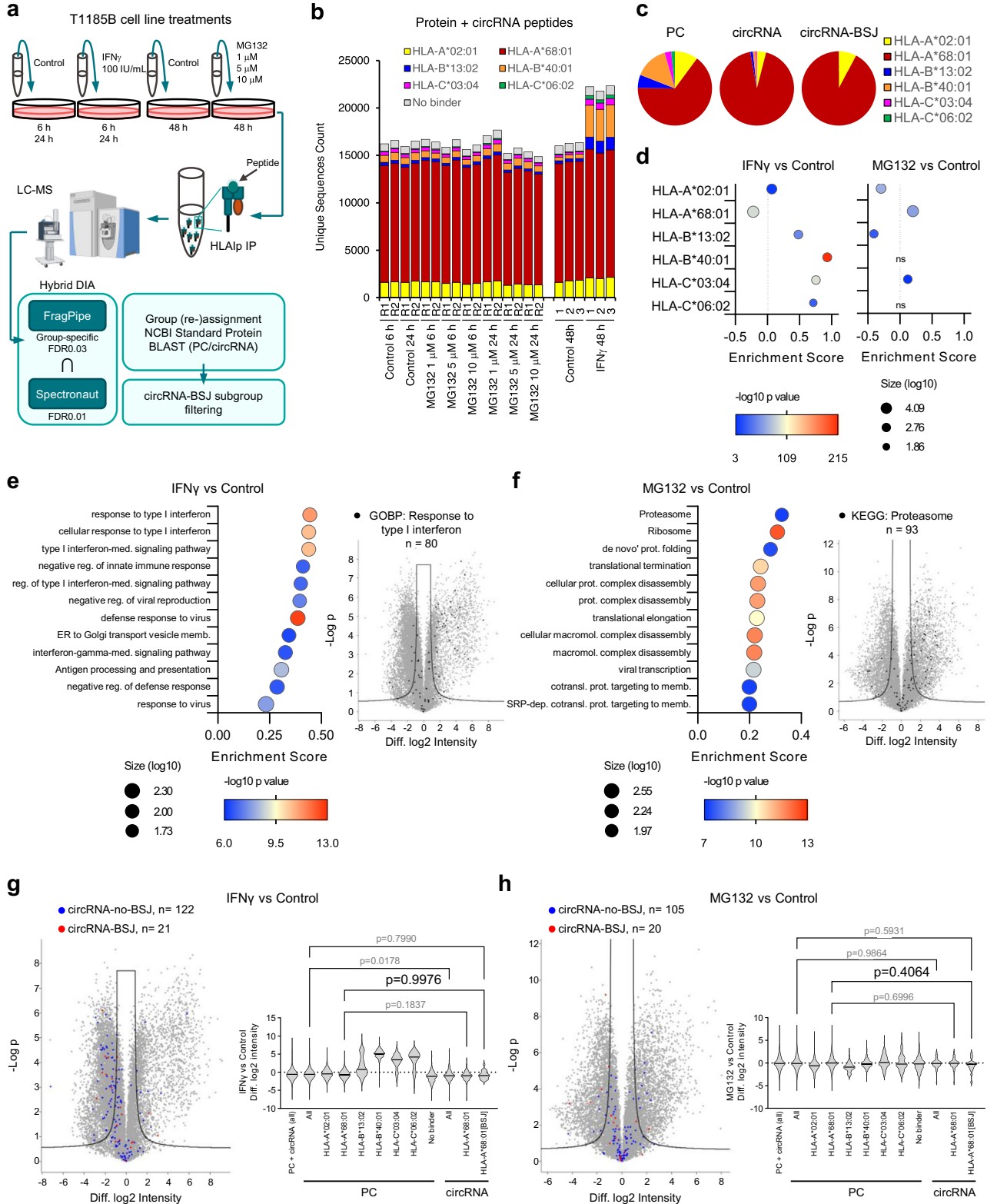

Interestingly, the peptide ILDKKVE[KL] was detected in five of the lung cancer patients, exclusively in tumor tissues, and it was not identified in any benign tissue included in the HLA Ligand Atlas[64]. Interestingly, the PC ILDKKVEKV peptide counterpart, that differs by only one amino acid, is ubiquitous and was detected in both tumor and healthy tissues. Both peptides are predicted to be strong binders of HLA-A*02:02 allele molecule with a % rank of 0.029 and 0.020 (NetMHCpan4.1), respectively. Since T cells that bind to ILDKKVEKV within the HLA-A*02:01

context were observed in the human T-cell repertoire[65], the potential immunogenicity of the ILDKKVE[KL] peptide becomes a compelling subject for further investigation. Furthermore, we searched the paired HLA-II immunopeptidomics data using the same workflow and identified overall 59,063 peptide sequences, ranging from 12,377–23,881 peptides across the patient's tumors and 9083–19,263 peptides in the adjacent healthy tissues. Nevertheless, only two circRNA-BSJ HLA-II peptides were identified (Supplementary Fig. 9 and Supplementary

**Fig. 4 | IFNγ or MG132 treatments similarly impacted the presentation of circRNA-derived and PC peptides in T1185B cells. a** The immunopeptidomics experimental design for the MG132 and IFNγ treatments. Peptide identification was done by hybrid DIA analyses using FragPipe and Spectronaut MS tools and a concatenated UniProt and trimmed circRNA-derived ORFs around the BSJ fasta file. Schema was partially created with BioRender.com. **b** Number of unique peptides identified by both tools in each sample, colored based on their predicted HLA restriction. **c** PC, circRNA, and circRNA-BSJ derived peptides predicted by NetMHCpan4.1 to bind (% rank< 2) the respective HLA-I alleles (best % rank score). **d** Enrichment analysis comparing peptide intensities in IFNγ versus Control (left) and MG132 versus Control (right) treatments, annotated based on their HLA restriction (two-sided Wilcoxon-Mann-Whitney test, 1D annotation enrichment). **e** GOBP, GOCC and KEGG enrichment analysis (left) of differentially presented peptides upon IFNγ treatment (two-sided Wilcoxon-Mann-Whitney test, 1D annotation enrichment, peptide number≥50; Benj. Hoch. FDR < 1E-04; top 12 categories). A volcano plot depicting differential presentation analysis (right) of the

effect of IFNγ on the immunopeptidomes, highlighting the most enriched GOBP category (Student's t-test FDR0.01, s0 = 0.75). **f** Enrichment analysis (left) of differentially presented peptides upon MG132 treatment (two-sided Wilcoxon-Mann-Whitney test, 1D annotation enrichment, Benj. Hoch. FDR < 1E-04; top 12 categories). A volcano plot depicting differential presentation analysis (right) of the effect of MG132 on the immunopeptidomes, highlighting the most enriched KEGG category (Student's t-test FDR0.01, s0 = 0.75). **g** Differential presentation of circRNA-derived peptides upon IFNγ treatment (Student's t-test FDR0.01, s0 = 0.75, left) and associated violin plot showing the difference in the log2 intensity of the peptides associated with their HLAs (one-way ANOVA and Sidak's multiple comparisons test, right). Due to the overrepresentation of HLA-A*68:01 in circRNA and circRNA-BSJ groups shown in c, only this HLA allele is represented for circRNAs. **h** Same as (**g**), for the MG132 treatment. In the volcano plots, peptides derived from proteins annotated as response to type I interferon or proteasome are highlighted in black, while circRNA-no-BSJ peptides and circRNA-BSJ are highlighted in blue and red, respectively. Source data are provided as a Source Data file.

Data 5), both predicted to be binders (NetMHCIIpan4.1, Supplementary Data 6), in agreement with the previous report that demonstrated very low contribution of non-canonical sources to the HLA-II immunopeptidome[63].

## Discussion

The human translatome potentially contains many undiscovered ORFs originated from the various translation frames of the linear transcripts, as revealed by the discovery of thousands of novel unannotated open reading frames (nuORFs) which populate the immunopeptidome[9]. Likewise, the expansion of the cancer related translatome repertoire to circRNA-derived ORFs, and especially those spanning the BSJ region represents an additional source of neoantigens that can be presented and potentially improve current immunotherapies. In the current study we developed a workflow to specifically identify HLA-presented circRNA-derived peptides, directing our MS search to peptides derived from predicted ORFs with a canonical initiation start codon and encoded within the region overlapping the BSJ. Our approach of considering only a flanking region with a maximum of 24 amino acids around the BSJ, reduces the search space while allowing the identification of 25-mer peptides having at least one amino acid spanning the BSJ. Thus, the BSJ-focused circRNA reference is also suitable for HLA-II immunopeptidomics studies.

With the MS-based immunopeptidomics approach, we identified circRNA-derived peptides spanning the BSJ region in two melanoma samples and in a cohort of eight multi-region lung cancer tissues. In contrast to previous studies[44,45], we used a stringent and standardized immunopeptidomics workflow by combining different MS search engines and/or applying a group-specific FDR to decrease the number of false identifications. Treatment of T1185B melanoma cells with IFNγ or the proteasome inhibitor MG132 did not alter differently the presentation of PC and circRNA-BSJ derived peptides, suggesting they follow similar routes of antigen processing and presentation. IFNγ treatment induced the upregulation of HLA expression, mainly the HLA-B*40:01, and induced the presentation of IFNγ regulated genes that are often massively upregulated following treatment[66]. Most circRNA-derived peptides we detected in T1185B cells are predicted to bind HLA-A*68:01. In general, their normalized presentation level is comparable to that of the canonical HLA-A*68:01 peptidome, but there is a tendency for lower presentation after treatment. The limited expression of their source proteins could be a factor that impedes their presentation, even when there is an upregulation of HLA expression. Furthermore, proteasome degradation is a main source of peptides for HLA presentation. Therefore, we explored the impact of proteasome inhibition on the presentation of canonical and circRNA-derived peptides in T1185B cells treated with the reversible proteasome inhibitor MG132. The degradation of dysfunctional proteasomes through autophagy is a known phenomenon[67,68] that could potentially result in

the increased presentation of the degradation products we observed in the treated cells. Moreover, a positive feedback loop might result in increased expression, albeit at substoichiometric levels, of certain proteasome subunits. Due to misfolding, these subunits could be swiftly degraded, potentially contributing to the immunopeptidome through this mechanism. In our experimental setup, the immunopeptidome derived from circRNAs exhibited a presentation profile that closely resembled the canonical peptidome. This observation suggests that it shares a similar dependence on the proteasomal pathway.

Cancer-specific translation of circRNAs and associated antigen presentation can lead to new therapeutic targets for tumor-immunotherapy. The existence of cancer-specific circRNAs has been shown at the transcriptional level[69]. Interestingly, QKI is upregulated during epithelial-mesenchymal transition (EMT) and boosts the abundance of circRNAs[20,70]. In our study we addressed cancer-specificity by analyzing HLA-I immunopeptidomic data derived from a cohort of eight lung cancer patients, comprising tumor and matched healthy tissues. Despite the limited number of patients and the absence of circRNA expression data, we identified a few HLA-presented circRNA-derived peptides exclusively detected in cancer tissues. For example, the circRNA-BSJ derived peptide ILDKKVE[KL] is predicted to be encoded within the *hsa_circ_0076651* in an open reading frame lacking stop codons through rolling circle translation. This circRNA is hosted by the *HSP90AB1* gene which expression is known to be upregulated in lung cancer and linked with poor overall survival after surgery[71]. Remarkably, while peptides derived from HSP90AB1 are often detected in the human immunopeptidome, it has been demonstrated that T cells that bind to ILDKKVEKV presented by HLA-A*02:01 molecules were observed in the human T-cell repertoire and were activated following viral infection[65]. As the circRNA-BSJ ILDKKVE[KL] peptide was identified solely within tumor tissues of five lung cancer patients, the exploration of its potential immunogenicity emerges as a compelling avenue for additional research. Beyond those circRNAs with cancer restricted expression, cancer-specific translation of circRNAs could potentially expand the repertoire of immunopeptides that are uniquely presented in cancer[72]. Hence, circRNA-derived peptides not overlapping the BSJ might also be of interest, as they represent a potential source of translated non-canonical peptides.

In the present work, we based our circRNA space search on the circBase database, while other circRNA public databases, such as riboCirc[43], TransCirc[39] and CSCD2[69] could also be used to guide the discovery of circRNA-derived peptides. Moreover, the large search space, which negatively affects the sensitivity of MS search engines[73], could be further adapted using prior knowledge about circRNA expression in a specific experimental setting. Newly discovered circRNA sequences obtained through capture sequencing[74] and nanopore sequencing[75] could be used as input for the generation of a

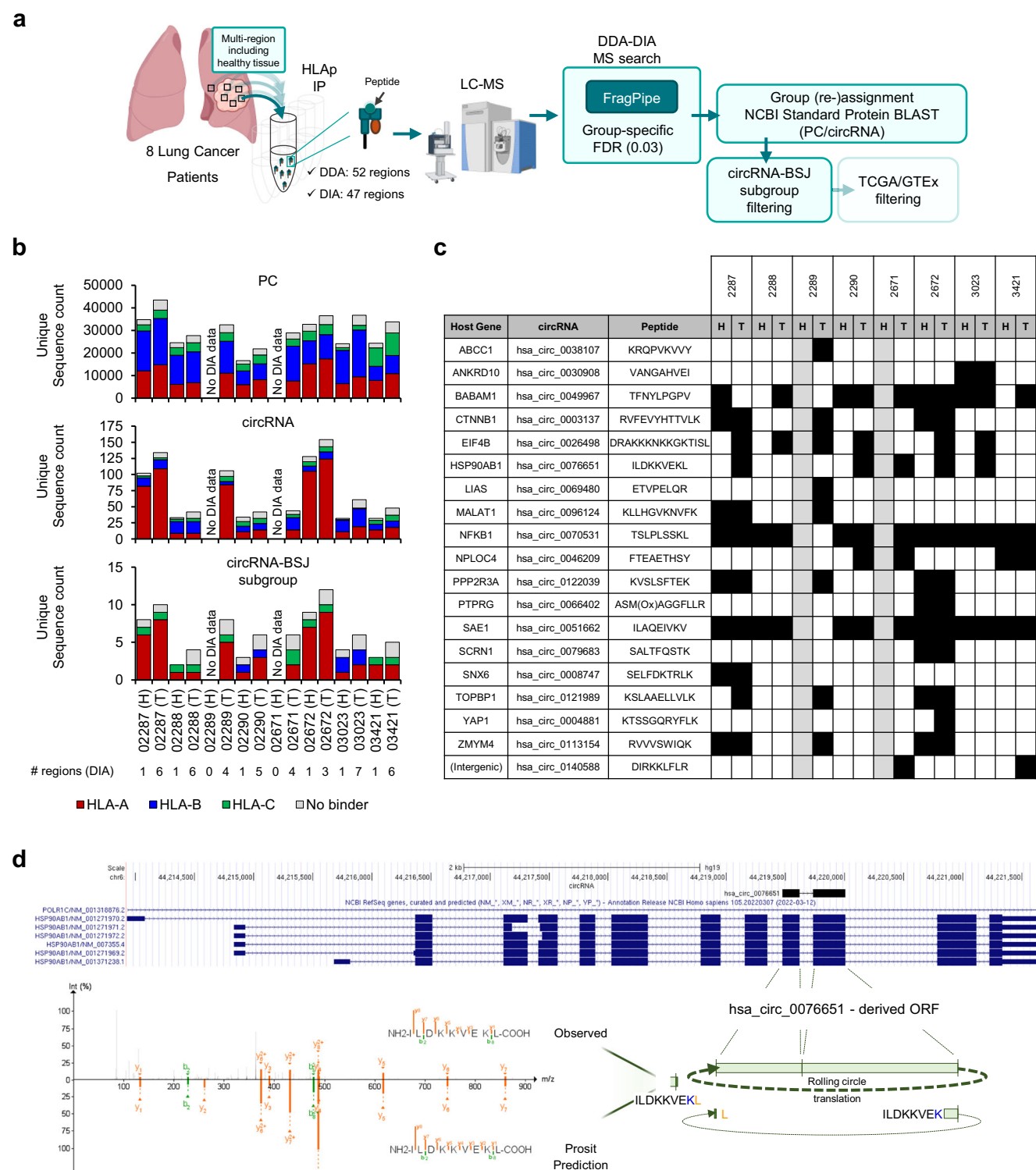

**Fig. 5 | Identification of potentially lung cancer-associated circRNA-BSJ peptides. a** Schema of the workflow used to explore immunopeptidomics DDA and DIA MS data of 8 lung cancer patients, including 52 tumoral and healthy matched tissues. MS raw files were analyzed by a DDA-DIA approach with FragPipe, applying a group-specific FDR threshold of 0.03 for protein-derived and circRNA-derived peptides groups. circRNA-derived peptides were blasted against a larger Reference Proteins database to re-assign matching peptides to the protein (PC) group. Schema was partially created with BioRender.com. **b** Count of unique PC, circRNA and circRNA-BSJ derived peptides per patient and tissue type, colored based on their predicted HLA restriction. Number of regions from where the peptides were identified (DIA data) are shown for each patient/tissue type. T: tumor; H: healthy. **c** Diagram showing the detection across patients of circRNA-derived peptide and regions where the peptides were identified. Black square: detected peptide; grey square: no data. **d** UCSC genome browser (hg19) visualization of *hsa_circ_0076651*, hosted by *HSP90AB1* gene (top), which is predicted to have rolling circle translation and encode the circRNA-specific peptide ILDKKVE[KL] across the BSJ. Mirror plot of a representative spectrum of peptide ILDKKVE[KL] measured by DDA (top) and the MS/MS predicted by Prosit (bottom), created by the PDV visualization tool with a fragment m/z tolerance of 10 ppm. Source data are provided as a Source Data file.

**Table 2 | HLA-I circRNA-derived peptides, overlapping the BSJ encoding region, detected in the lung cohort (eight patients, six with paired healthy tissue) through DDA-DIA FragPipe analysis with group-specific FDR calculation**

| Gene ID | circRNAs_frame | Infinite translation candidate | Peptide Sequence | Length (AA) | NetMHCpan – 4.1 Cohort Binder | BSJ context (3'Exon -> Peptide [BSJ] <- 5'Exon) | Healthy detection (#Patients) | Tumor detection (#Patients) | circRNA TCGA/GTEx pass filter |
|---|---|---|---|---|---|---|---|---|---|
| ABCC1 | hsa_circ_0038107_1 | FALSE | KRQPVKVVY | 9 | TRUE | MSRSMGE -> K[R]QPVKVVY <- SSKDPAQPKESSKVDAN | 0 | 1 | FALSE |
| ANKRD10 | hsa_circ_0030908_0 | FALSE | VANGAHVEI | 9 | TRUE | PIHKAARSGSLECISAL -> VANGAHV[E]I <- VRVKLPFTRQLALGA | 1 | 1 | FALSE |
| BABAM1 | hsa_circ_0049967_0 | FALSE | TFNYLPGPV | 9 | FALSE | GLTSDPRELCSCLYDLETASCS -> TF[N]YLPGPV <- RGNVTAKAGVVQRLQNQR | 4 | 6 | FALSE |
| CTNNB1 | hsa_circ_0003137_1 | FALSE | RVFEVYHTTVLK | 12 | TRUE | MATKKA -> [R]VFEVYHTTVLK <- IQRGQWLLKLI | 2 | 3 | FALSE |
|  | hsa_circ_0004030_1 | FALSE |  |  |  | SWMGCLQVTAISWPGLILTCKSSF -> [R]VFEVYHTTVLK <-IQRGQWLLKLI |  |  |  |
| EIF4B | hsa_circ_0026498_2 | TRUE | DRAKKKNKKGKTISL | 15 | FALSE | DRRDDRSWSSRDDYSRDDYRRD -> DR[A]KKKNKKGKTISL <- TDFLAEDGGTGG | 0 | 6 | FALSE |
| HSP90AB1 | hsa_circ_0076651_0 | TRUE | ILDKKVEKL | 9 | TRUE | EESKAKFENLCKLMKE -> ILDKKVE[KL] <- GIHEDSTNRRRLSELLRYHTSGS | 0 | 5 | FALSE |
| LIAS | hsa_circ_0069480_1 | FALSE | ETVPELQR | 8 | FALSE | AIEKVALSGLDVYAHNV -> ETVPELQ[R] <- YA | 0 | 1 | FALSE |
| MALAT1 | hsa_circ_0096124_2 | FALSE | KLLHGVKNVFK | 11 | TRUE | M-> [KL]LHGVKNVFK <- RKLRERTTEPRINT | 1 | 2 | FALSE |
| NFKB1 | hsa_circ_0070531_1 | FALSE | TSLPLSSKL | 9 | TRUE | MIC-> T[S]LPLSSKL <- QSIKILILQNQPLCLSS | 6 | 7 | FALSE |
| NPLOC4 | hsa_circ_0046209_1 | FALSE | FTEAETHSY | 9 | TRUE | WWRMRLISTSANRTGR-> FTEAETHS[Y] <- NSCPVPGWSEADHSNKERNSSNIF | 1 | 3 | FALSE |
| PPP2R3A | hsa_circ_0122039_2 | FALSE | KVSLSFTEK | 9 | TRUE | SPVGDKAKDTTSAVLIQQTPEVI-> [KV]SLSFTEK <- | 2 | 3 | FALSE |
| PTPRG | hsa_circ_0066402_0 | FALSE | ASM(Ox)AGGFLLR | 10 | TRUE | MAQRALNT-> ASMAGGFLL[R] <- HVTSCMEAWPVSSPALFVPSQPAP | 1 | 1 | FALSE |
|  | hsa_circ_0066406_0 | FALSE |  |  |  | MAQRALNT-> ASMAGGFLL[R] <- CLWS |  |  |  |
|  | hsa_circ_0124401_0 | FALSE |  |  |  | MAQRALNT-> ASMAGGFLL[R] <- YYRTCSD |  |  |  |
| SAE1 | hsa_circ_0051662_2 | FALSE | ILAQEIVKV | 9 | TRUE | MAPVCAVVGG-> ILAQEIV[KV] <- LLLRDGPSVCGGWRDFGTGNCEG | 6 | 7 | FALSE |
| SCRN1 | hsa_circ_0079683_0 | FALSE | SALTFQSTK | 9 | TRUE | GVSVLPQNRSSPCIHYFTGTPDPS-> [S]ALTFQSTK <- FQGPMP | 1 | 1 | FALSE |
| SNX6 | hsa_circ_0008747_0 | TRUE | SELFDKTRLK | 10 | TRUE | LGTQDSTDICKFFLKV-> SELFDKT[RL]K <- AINVDLQSDAALQVDISDALSE | 1 | 1 | FALSE |
| TOPBP1 | hsa_circ_0121989_1 | FALSE | KSLAAELLVLK | 11 | TRUE | MYTPHCART-> K[S]LAAELLVLK <- | 1 | 3 | FALSE |
| YAP1 | hsa_circ_0004881_0 | FALSE | KTSSGQRYFLK | 11 | TRUE | MA-> KTSSGQRYFL[K] <- PVLMQALQEP | 0 | 1 | FALSE |
| ZMYM4 | hsa_circ_0113154_2 | FALSE | RVVVSWIQK | 9 | TRUE | DKAANQVEETLHTHLPQTPETNF-> [RV]VVSWIQK <- CLKI | 2 | 3 | FALSE |
| (Intergenic) | hsa_circ_0140588_0 | FALSE | DIRKKLFLR | 9 | FALSE | LGVCLVSLRWWRHACLPNMW-> DIR[KK]LFLR <- N | 0 | 2 | FALSE |

Amino acids spanning the BSJ are represented within squared brackets and bold font. Prediction of best binder allele molecule among patients where peptides were detected by MS is detailed in Supplementary Data 4.

sample-specific circRNA reference file to potentially identify sample-specific circRNA-derived peptides. Alternatively, in the absence of circRNA expression data, the expression of the linear counterparts might be used to restrict the search space to circRNAs from the same host genes, assuming their co-existence[25,76]. In conclusion, we have established a dedicated workflow to identify HLA-presented circRNA-derived peptides. We exemplified its utility in identifying cancer-related peptides that can be further explored as candidates for immunotherapy. Our approach is versatile, and candidate peptides from circRNAs can be investigated in various biological and pathological contexts.

## Methods

### Patient samples, cell lines and cell culture

An informed consent was given by the participants, according to the requirements of the institutional review board (Ethics Commission, Centre hospitalier universitaire Vaudois, CHUV). Cell line T1185B was derived from non-lymphoid metastasis of a melanoma patient at the Ludwig Institute for Cancer Research, Department of Oncology, University of Lausanne[77]. Cell line was grown in RPMI 1640 Medium GlutaMAX™ Supplement (Gibco) with 0.55 mM L-arginine (Sigma), 0.24 mM L-asparagine (Sigma), 1.5 mM L-Glutamine (Gibco), 10 mM HEPES (Gibco), 10% of heat-inactivated fetal bovine serum (FBS), 100 U/mL penicillin and 100 µg/mL streptomycin (BioConcept). Snap-frozen tumor tissues from different regions of a lymph node of Mel-1 melanoma patient (clinical study: NCT03475134) were collected and stored at −80 °C. A cell line was generated from the same patient's tumor at the CTE Biobank (CHUV) and grown in RPMI 1640 Medium GlutaMAX™ Supplement (Cat# 61870010, Gibco) with 10% of non-heat inactivated FBS, 100 U/mL penicillin and 100 µg/mL streptomycin. After in vitro expansion, both cell lines were trypsinized, washed twice in PBS and dry pellets containing $1 \times 10^8$–$2 \times 10^8$ cells were collected and stored at −80 °C, before HLA-I immunoprecipitation (HLA-IP) workflow.

### Treatment with the proteasome inhibitor MG132 and IFNγ

T1185B cells were treated with different concentrations of MG132 (S2619, Selleckchem): 1 µM, 5 µM and 10 µM or DMSO (MG132 vehicle) for 6 h and 24 h, and with 100 IU/mL IFNγ (130-096-484, Miltenyi Biotec) or water (IFNγ vehicle) for 48 h. After treatments, cells were harvested, and cell pellets were collected for HLA-IP. After purification, immunopeptides from each condition of the MG132 treatment were measured by MS in technical duplicates. Three biological replicates were used in each condition of the IFNγ treatment (one MS injection each).

### Purification of HLA-I peptides and LC-MS/MS analysis

HLA-I immunoprecipitation was performed using the Waters Positive Pressure-96 Processor (Waters, Milford, MA)[6,66] and the number of samples and replicates are indicated in Supplementary Data 7. Shortly, protein-A Sepharose 4B (Pro-A) beads (Invitrogen) were used to purify W6/32 monoclonal antibodies from the supernatant of HB95 hybridoma cells (ATCC HB-95). After antibody crosslinking, Pro-A beads were used for immunoaffinity purification of HLA-I complexes from tissue or cell line lysates. HLA-I peptides were then purified using a C18 solid phase extraction (SPE) and dried using vacuum centrifugation (Concentrator plus, Eppendorf). Samples were stored at −80 °C if not immediately submitted to mass spectrometry analysis. Finally, immunopeptides were re-suspended in 2% ACN and 0.1% FA (formic acid). iRT peptides (Biognosis, Schlieren, Switzerland) were spiked into in the samples as indicated in Supplementary Data 7 and analyzed by LC-MS/MS.

### Liquid chromatography and mass spectrometry (LC-MS)

The LC-MS system consisted of an Easy-nLC 1200 coupled to Q Exactive HF-X mass spectrometer (ThermoFisher scientific, Bremen,

Germany) or to Eclipse tribrid mass spectrometer (ThermoFisher Scientific, San Jose, USA). The peptides were eluted on a 450 mm analytical column (8 µm tip, 75 µm ID) packed with ReproSil-Pur C18 (1.9 µm particle size, 120 A pore size, Dr Maisch, GmbH) and separated at a flow rate of 250 nL/min as described in ref. 66. For DDA measurements, the top 20 most abundant precursor ions selection was performed on the Q Exactive as described[66]. For DIA, the Eclipse tribrid mass spectrometer was used to sample ions. The cycle of acquisitions consists of a full MS scan from 300 to 1650 $m/z$ (R = 120,000, ion accumulation time of 60 ms and normalized AGC of 250%) and 22 DIA MS/MS scans in the orbitrap. For each DIA MS/MS scan, a resolution of 30,000, a normalized AGC of 250%, and a stepped normalized collision energy (27, 30, and 32) were used. The maximum ion accumulation was set to auto, the fixed first mass was set to 200 $m/z$, and the overlap between consecutive MS/MS scans was 1 $m/z$ as described in ref. 78.

### Parallel reaction monitoring (PRM)

Synthetic heavy labelled peptides were ordered from Thermo Fisher Scientific as crude (PePotec grade 3). After re-suspension in 2% ACN in 0.1% FA, synthetic peptides were individually analyzed by MS to confirm the lack of contaminating light counterpart. In a second step, synthetic peptides were spiked into the endogenous immunopeptides (0.5 or 1 pmol µL⁻¹) and PRM was performed on both light (endogenous) and heavy peptides[6] (Supplementary Table 1). Collected data was analyzed using Skyline (64-bit, v20.2.0.343), using an ion mass tolerance of 0.05 $m/z$. The resulting MS/MS spectra were further sequenced with the mass spectral peak labeling tool pLabel™ (v2.4.3)[79,80]. Following manual inspection of the results, peptides were annotated as PRM-validated (+) and PRM non-validated (-). Other tested peptides were defined as 'failed QC' (due to too long elution profile), and as 'inconclusive' due to noisy signal (Table 1).

### Construction of reference FASTA files for the identification of non-canonical circRNA peptides by mass spectrometry

To explore the contribution of circRNA sources to the immunopeptidome, we first generated a reference file of circRNA sequences present in the circBase database (http://www.circbase.org/)[49]. circBase holds more than 140,000 circRNAs from exonic, intronic and intergenic loci, from both coding and non-coding regions, represented by linear sequences. The BSJ and its sequence context was created for each circRNA by joining the 3' end to the 5' start of the linear sequence. For illustration purposes, Fig. 1a, b refer to circRNAs composed solely by exons, but the strategy was applied to all putative spliced circRNA sequences from circBase. To facilitate the in silico translation of the BSJ-containing circRNA fragments, we concatenated four copies of the circRNA sequence into a single sequence, termed "4x circRNA" in which the circRNA reading frame changes at each of the three internal BSJs (Supplementary Fig. 1a). In this way, all the possible BSJ reading frame transitions are covered in a single in silico translation of the "4x circRNA" sequence. In circRNA sequences that contain an integral number of codons, the reading frame does not change at the BSJ, so only two concatenated circRNA sequences are required, with each invariable reading frame initiated at nucleotide position 1, 2 or 3 in the "2x circRNA" concatenated sequence (Supplementary Fig. 1b). For each 4x (or 2x) circRNA sequence we assembled a list of the sequence-based nucleotide coordinates specifying all ATG codons, the codon containing each BSJ and all stop codons. This list of elements, sorted by coordinate order, was then traversed to isolate one or more BSJ elements bounded by stops (eg. STP-2350, ATG-2368, BSJ-2722, STP-2770) that have the potential to be translated. These "stop-to-stop" sequences were in silico translated into peptide sequences (excluding the stop codons), and the amino acids N-terminal to the first methionine residue were removed. Stop-to-stop elements that did not contain an ATG upstream the BSJ were discarded. Where possible, peptides were further truncated such that the final sequence contained 24

amino acids flanking the BSJ-containing residue (or 23 amino acids if the BSJ was located between two codons). This made the sequences suitable for both HLA-I and HLA-II MS-based immunopeptidomics workflows. The resulting circRNA-derived BSJ-ORF fasta file contains sequences with up to 49 amino acids covering the transcript position corresponding to at least one BSJ (24 amino acids upstream the BSJ, one amino acid partially encoded by the BSJ and 24 amino acids downstream of the BSJ). These sequences were concatenated with a human UniProt fasta file with isoforms [Reviewed (Swiss-Prot), 42362 entries, downloaded on 2022-03-07][51] before performing the MS database search. The concatenated fasta file was adapted for compatibility with the group-specific FDR calculation in FragPipe following the structure of UniProt fasta file headers[50,51]. Two groups were assigned through the attribution of a different number in the Protein Existence field of the fasta headers (PE). PE = 1 was attributed to the protein group (including common MS protein contaminants) and PE = 4 was attributed to the circRNA group, allowing FragPipe to annotate sequences to the different groups for the group-specific FDR calculation.

## Mass spectrometry database search workflow

DDA MS search with group-specific FDR: MS-derived raw files resultant from three biological replicates of T1185B (Supplementary Data 7) were searched using MaxQuant[52] (version 2.1.0.0) with a PSM FDR of 0.1 and Comet[53] against the generated reference fasta file containing both UniProt and the trimmed circRNA-derived putative ORFs around the BSJ and initiated by the canonical start codon ATG. Outputs were then intersected by NewAnce (v1.6) (https://github.com/bassanilab/NewAnce), setting a group-specific FDR of 0.03 for protein- and circRNA-derived peptides. The search was done setting a nonspecific protein digestion cleavage, no fixed modifications, methionine oxidation and protein N-term acetylation as variable modifications, and restricting the peptide length to 8–15 amino acids. Same approach was used for the cell line and tumor tissues of patient Mel-1 patient (two biological replicates and three different lymph node regions, respectively). NewAnce comprised a PDV format output which was used to visualize a representative spectrum/best PSM of the identified circRNA-derived peptides. PDV 1.7.4[55] was used with a tolerance of 10 ppm.

Hybrid DIA MS search: DIA files corresponding to the immunopeptidomics samples of MG132 and IFNγ treatment of T1185B cells were searched using a hybrid DIA approach using two computational tools, FragPipe (v.20.0) with group-specific FDR calculation[58,61,81] and Spectronaut (v.18.4)[82], against the generated reference fasta file containing both UniProt and the trimmed circRNA-derived ORFs around the BSJ to which a list of common MS contaminant proteins were added. To increase the coverage of the spectral library, we assembled available DDA raw files from T1185B cells treated or not with IFNγ (from the PRIDE accession PXD013649[6]), the newly generated DDA data, together with the DIA files of T1185B cells treated with MG132 and the new IFNγ treatments (and their respective controls), as indicated in Supplementary Data 7. In FragPipe we applied a group-specific FDR threshold of 0.03 (MSFragger Group variable: Protein evidence from FASTA file) while in Spectronaut we applied a global peptide FDR threshold of 0.01. In both engines, the search was done by applying a FDR threshold of 1 for proteins, nonspecific protein digestion cleavage, no fixed modifications, methionine oxidation and protein N-term acetylation as variable modifications, and restricting the peptide length to 8-15 amino acids. Hybrid spectral libraries were then used to match and quantify peptides from the immunopeptidomics DIA data using a peptide precursor group-specific FDR of 0.03 for FragPipe or global FDR of 0.01 for Spectronaut. Default decoy generation methods were used for each MS search tool, reversed and mutated sequences for FragPipe and Spectronaut, respectively. Data analysis was performed using Fragpipe quantification values after overlapping identified sequences from both FragPipe and Spectronaut MS analysis tools.

DDA-DIA MS search with group-specific FDR calculation: HLA-I and HLA-II raw files of the lung cancer cohort of 8 patients and 52 tumoral and healthy matched tissues[63] (Supplementary Data 7) were downloaded from PRIDE PXD034772 and analyzed by FragPipe (v.19.2-build39 for HLA-I and v.20.0 for HLA-II immunopeptidomes) which supported group-specific FDR calculation. Spectral library generation was performed using the DDA immunopeptidomics data. The search was done setting a nonspecific protein digestion cleavage, no fixed modifications, methionine oxidation and protein N-term acetylation as variable modifications, a group-specific FDR threshold of 0.03 for peptides (MSFragger Group variable: Protein evidence from FASTA file), a FDR threshold of 1 for proteins, and restricting the peptide length to 8-15 or to 8-25 amino acids for HLA-I or HLA-II MS searches, respectively. Respective spectral libraries were then used to match and quantify peptides from the immunopeptidomics DIA data using a peptide precursor group-specific FDR of 0.03. DIA immunopeptidomics raw files were used for matching and quantification of peptides. Peptides from canonical and non-canonical circRNA groups were used to calculate the FDR separately because the score distributions are different. Pooling them together would result in underestimated FDR for the circRNA group. HLA-I and HLA-II library generation and peptide identification were performed separately.

For MS/MS prediction, an ecosystem within Prosit[83] was used via Oktoberfest[84]. Oktoberfest can calibrate collision energy (CE), rescoring search results and generates spectral libraries from a list of peptides. We used "Prosit_2020_intensity_HCD" and "Prosit_2019_irt" as models for intensity and retention time predictions respectively. For peptide ILDKKVEKL prediction, we used CE = 28 for charge state z = 1 + , z = 2+ and z = 3 + . Comparison of the DDA-measured and the Prosit-predicted spectra of ILDKKVEKL was performed using PDV 1.8.2[55] with a tolerance of 10 ppm.

Furthermore, to remove potentially ambiguous identifications resulting from PSMs that better fit possible modified PC sequences, the MSMS spectra of candidate circRNA-BSJ peptides were re-searched with COMET (same parameters as above, but no FDR) against the human reference proteome UniProt database concatenated with the list of the circRNA-BSJ peptide sequences, including six common modifications. The variable modifications included were 15.9949 Da for oxidation on M, 42.010565 Da for acetylation on the N-terminus, 79.966331 Da for phosphorylation on STY, 119.004099 Da for cysteinylation, 0.98402 Da for deamidation NQ and 57.021464 Da for carbamidomethyl on C. Ambiguous identifications mapping to modified PC peptides with either higher or equal XCorr (delta score =0) were excluded from downstream analyses.

## Identification of peptide candidates uniquely derived from circRNAs

After MS search, common MS contaminants were removed and peptides matching both the protein and circRNA groups were annotated as belonging to the protein-coding space. In addition, circRNA-derived peptide candidates were blasted against the Reference Proteins database (refseq_protein, homo sapiens, 106 K entries) using the online NCBI Standard Protein Blast tool[85] and peptides with 100% similarity to sequences in this reference were re-annotated as belonging to the protein group. circRNA-derived peptides were further mapped to the original trimmed circRNA-derived ORF sequences to identify those overlapping the BSJ encoding region.

## Assessing tumor specificity

Cancer related circRNAs in the lung cancer cohort were defined as having expression of their host gene of at least 2.5 TPM at the 99th percentile in any TCGA cancer type but not higher than 1 TPM in any GTEx[86] normal tissue samples at the 90th percentile (excluding testis and sun exposed skin). We assumed the co-existence between circRNA

and linear counterparts, since the biogenesis of circRNAs compete with regular splicing[25].

In addition, we downloaded the MS files of HLA-I immunopeptidomics data from the HLA Ligand Atlas[64] and searched them against the fasta file containing both UniProt and the trimmed circRNA-derived putative ORFs around the BSJ and initiated by the canonical start codon ATG, to obtain information about their detection in benign tissues. We used Comet and the NewAnce tool, setting a nonspecific protein digestion cleavage, no fixed modifications, methionine oxidation as variable modification, a group-specific FDR threshold of 0.03 for PC and circRNA-derived peptides, a FDR threshold of 1 for proteins, and restricting the peptide length to 8-15 amino acids.

## HLA typing
Genomic DNA from T1185B cell line and Mel-1 tumor tissue was extracted with the DNeasy Blood & Tissue Kit (Qiagen). HLA-typing was performed using the TruSight HLA v.2 Sequencing Panel protocol (CareDx). Sequencing was performed on an Illumina® MiniSeq™ System (Illumina) and data was analyzed using the Assign TruSight HLA v.2.1 software (CareDx).

## Prediction of HLA binding affinity
NetMHCpan4.1[87] and NetMHCIIpan4.1[88] were used to predict the binding affinity of the identified peptides against the patient-specific HLA-I allotypes. Peptide sequences with a % rank lower than 2 were considered as binders. Each binder peptide was then annotated to the best binder allele considering the lowest % rank.

## circRNA detection and Sanger sequencing
RNA was extracted with the miRNeasy Tissue/Cells Advanced Mini Kit (Qiagen) according to the manufacturer's protocol. Purification of total RNA transcripts include a non-enzymatic gDNA removal by a gDNA eliminator column. RNA quality was checked on a Fragment Analyzer using the RNA kit (DNF-471-0500, Agilent Technologies). Total RNA was converted into cDNA using the GoScript™ Reverse Transcriptase kit with random primer hexamers (A2801, Promega). Amplification of the cDNA (equivalent to 100 ng of total RNA) flanking the circRNA BSJ, was performed by a divergent RT-PCR, using oligos designed outside the BSJ, allowing validation of the BJS with Sanger Sequencing (Supplementary Table 2). RT-PCR amplification was performed using the KAPA HiFi HotStart ReadyMix PCR Kit (7958935001, Roche) using manufacturer's recommendations, except for the number of cycles which was increased to 40 to increment the amount of product to be sequenced. RT-PCR products were run in a 2% agarose gel, bands with the expected size were excised and DNA purified using the NucleoSpin Gel and PCR clean-up kit. Direct Sanger Sequencing of the purified DNA was performed at Microsynth (Balgach, Switzerland).

## Data analysis and statistics
Results are shown for peptides of 8-15 and 9-25 amino acids in length for HLA-I or HLA-II datasets, respectively. Statistical analysis of the hybrid DIA MS search addressing MG132 and IFNγ treatments were performed by Perseus 1.5.5.3[89], after summing the intensities of all precursor charge states for all peptides identified in FragPipe, keeping unmodified and modified peptides separately. Canonical source proteins were annotated with GOBP, GOCC, KEGG and keyword annotations. Common MS contaminants were removed, and the output was split in two datasets analyzed in separate, each one containing only the peptides identified in MG132 or IFNγ treatments with their respective controls. Peptide intensities from FragPipe were log2 transformed and the missing values were imputed from a normal distribution (width=0.3, down shift=1.8), followed by a width adjustment normalization and keeping peptides which sequences were also detected in the Spectronaut MS search, independently on their modifications. For standard hierarchical clustering the intensities of each peptide were z-scored. Otherwise, the log2 intensity matrix was further filtered to retain peptides identified and quantified in at least 80% of the raw files of the associated treatment. Raw files derived from the MG132 treatment were annotated in different categories based on the previous generated hierarchical clustering, Control (6 h and 24 h) and MG132 at 24 h (1 μM, 5 μM and 10 μM at 24 h), while for the IFNγ treatment we annotated the raw files as Control or IFNγ (48 h). The created groups were used to generate volcano plots for each treatment (s0 = 0.75; FDR0.01). The overall effect of the different treatments in the log2 intensity of the identified peptides was checked by performing a two-sample test, using a Student's t-test with s0 = 0.75 and a permutation-based FDR of 0.01. The Student's t-test difference between the different groups (per treatment group) were then used to calculate the peptide presentation enrichment score per GOBP, GOCC and KEGG term, and per keyword annotation with 1D annotation enrichment analysis of the associated protein annotations with a Benjamin-Hochberg FDR threshold of 1E-04. The peptide presentation enrichment score per peptide group (PC and circRNAs) and HLA restriction was estimated using a less stringent FDR of 0.02 (Supplementary Data 2 and Supplementary Data 3). GraphPad Prism (version 9.5.1) was used to create a bubble plot representing the enrichment of peptides in each HLA allele and GOBP, GOCC and KEGG categories (peptide number≥ 50, top 12 categories with the higher enrichment score; keyword categories were excluded in this representation). Violin plots showing the difference in the log2 intensity of the peptides associated with their HLA restriction were also generated using GraphPad. DeepVenn (https://www.deepvenn.com) was used to visualize the intersection between FragPipe and Spectronaut. Upset plots showing the intersection of the identified peptide sequences in the three biological replicates of T1185B, annotated in the different PC, circRNA, and circRNA-BSJ groups were generated using Intervene (https://intervene.shinyapps.io/intervene/)[90].

## Reporting summary
Further information on research design is available in the Nature Portfolio Reporting Summary linked to this article.

## Data availability
MS raw files, reference fasta files and NewAnce, FragPipe and Spectronaut parameters and outputs generated in this study have been deposited in the ProteomeXchange Consortium via the PRIDE partner repository[91] under accession code PXD043989. The lung cancer cohort MS data used in this study are available in the ProteomeXchange Consortium under accession code PXD034772. Publicly available DDA raw files derived from T1185B cells treated or not with IFNγ used in this study are available in the ProteomeXchange Consortium under accession code PXD013649. The circBase database can be found at http://www.circbase.org/. The UniProt database can be found at https://www.uniprot.org. The Cancer Genome Atlas (TCGA) data can be found at https://www.cancer.gov/tcga. GTEx Portal can be accessed through https://www.gtexportal.org/home/[86]. HLA Ligand Atlas can be accessed through https://hla-ligand-atlas.org/welcome. The Human Protein Atlas can be accessed through proteinatlas.org. Source data are provided with this paper.

## Code availability
The provided GitHub link contains the necessary code to produce the circRNA-derived BSJ-ORF fasta reference and modify it to align with FragPipe's group-specific FDR calculation. https://github.com/bassanilab/CircRNA_MS_ref_fasta[50].

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

## Acknowledgements

We are thankful to Katja Muehlethaler for the generation of the primary cell line Mel-1 and further technical support, and to Anne-Christine Thierry, Petra Baumgaertner, Noemie Fahr, Aymeric Auger, Julien Schmidt, Philippe Guillaume and Alexandre Harari, from the Department of Oncology UNIL CHUV, for their contributions to the discussion of the manuscript. We are thankful to Gabriel Villamil and Uwe Ohler, from The Berlin Institute for Medical Systems Biology, Max Delbruck Center for Molecular Medicine, for their supportive discussions. This study was supported by the Ludwig Institute for Cancer Research, by the Swiss Cancer Research Foundation, grant KFS-4680-02-2019 (M.B.-S.) and

the Swiss National Science Foundation, PRIMA grant PR00P3_193079 (M.B.-S.). Some elements from Fig. 5 and Supplementary Fig. 9 were originally created using BioRender.com.

## Author contributions

H.J.F. and M.B.-S. conceived and designed the project and interpreted the results. B.J.S. constructed the circRNA fasta reference and interpreted the results. H.S.P. assisted in the LC-MS experiments (DDA, DIA, PRM) and MS search analysis. M.M. developed and implemented the software NewAnce for group-specific FDR calculations. L.E.K. collected study material. H.J.F., J.A.O., J.M., and M.T.-C. conducted immunopeptidomics MS experiments. F.Y., and A.I.N implemented the group-specific FDR in FragPipe. F.H., E.R.-A., A.I.K., and D.E.S. assisted in data interpretation. H.J.F. and M.B.-S. wrote the manuscript with contributions from all authors.

## Competing interests

The authors declare no competing interests.
