## [Peer Review File · Nature Communications]

REVIEWER COMMENTS

Reviewer #1 (Remarks to the Author):

Humberto Ferreira and colleagues suggest a new methodology and pipeline for the identification of unique circRNA-derived immunopeptides to expand the available pool of tumor-specific antigens and epitopes to further improve anti-cancer immunotherapy. The Bassani-Sternberg lab has previously impressively showcased ways to identify non-canonical immunopeptides from non-coding regions and further their approach with the current work focusing on translated circRNAs. The authors describe how a reference fasta file of circRNA sequences from circBase is created, followed by “stop-to-stop” circRNA fragment identification and in silico translation given the presence of the translation initiation codon ATG upstream of the back-splice junction (BSJ). Peptides were further truncated when applicable to 49 amino acid stretches, in which the central amino acid was at least partially encoded by the BSJ sandwiched by 24 amino acids up- and downstream of the BSJ. The particular lengths were chosen so that also longer MHC class II ligands, with a maximum length of 25 amino acids, might be identified by this approach in the future. The authors use multiple software tools or group-specific FDR to reduce the number of false positive circRNA-derive immunopeptide identifications. They employ targeted MS and Sanger sequencing to confirm their findings and identify potential tumor-specific circRNA-derived immunopeptides with potential for therapeutic intervention.

The manuscript is very well written and provides a very valuable contribution to further expand on the targetable tumor antigenic pool. Two earlier studies have attempted to identify circRNA-specific immunopeptide sequences (as cited by the authors), but due to methodological limitations have not convincingly shown the identifications of immunopeptides unique to translated circRNA. The current manuscript therefore presents the first fully validated attempt to identify MHC class I-presented immunopeptides from translated circRNA. All references included are timely and support the stated points. All relevant data has been uploaded to the PRIDE archive or is already publicly available on PRIDE from previous publications. While the manuscript is of high relevance to the fields of immunopeptidomics, antigen processing and presentation as well as cancer immunotherapy, a number of mostly minor points should be addressed before the manuscript is deemed suitable for publication in the highly esteemed journal Nature Communications.

1. If the reviewer understood the circRNA fasta file generation process correctly, only circRNAs solely composed of exonic sequenes are considered for the presented workflow. It would be interesting to know which fraction of all translated circRNAs this corresponds to? If a substantial fraction of translated circRNA is missed by the workflow, the authors might consider discussing a future extension of their pipeline to cover all translated circRNAs.

2. An additional figure in the main manuscript about the number and characterization of the identified canonical and circRNA-derived peptides would be appreciated. Even though tables 1 and 2 provide some of this information already, the reviewer believes that it is important to display an overview in peptide length, peptide motifs and binding affinities in the main text to instill confidence into the obtained data, which is available already in the supplements, but should be presented more prominently.

3. The circRNA-derived peptides seem to be slightly longer than the canonical peptides, particularly when focusing on the BSJ-spanning peptide sequences. Please discuss this in the first paragraph of the results or the discussion section. Could these longer peptides trigger a more specific T cell response due to looping out of the binding groove creating a more unique interaction surface for the TCR?

4. What could be the reason in your opinion for this interesting preference for a single HLA allele for circRNA-derived peptides? Are strong differences in expression levels the likely cause? What are the expression levels of the respective HLA alleles?

5. How many of the BSJ-spanning immunopeptides from table 1 were tested in total by PRM and how many of these could not be confirmed? Were the tested peptides that failed PRM QC of rather low search engine score and could potentially be false positives?

6. For this reviewer, it is hard to confirm from the current Volcano plots in panels 4f and 4h that the circRNA-derived BSJ-spanning peptides follow the same trend upon IFN γ and MG132 treatment as the canonical presented immunopeptides. Please consider a different graphical representation of the global up- and down-regulation of canonical and circRNA-derived immunopeptides to show this more conclusively and potentially even allow statistical testing.

7. LINE 80: Please add a brief sentence to explain the back-splice junction since some readers might not be familiar with it.

8. LINE 128: Add here the lengths of MHC class I and II peptides for readers to more easily see why 49 amino acids were chosen as peptide truncation length.

9. Tables 1 and 2: Please change the color-code of the peptide sequence and the 5' exon to facilitate readability/ease of information intake for red-green colorblind people.

10. Which peptides were found in both the DDA and DIA experiments? Please include a column to indicate this information in Table S1.
11. Figure 2: The annotations of b and y-ions are not legible in the spectra and the sequence above seems to be distorted. Please correct.
12. Figure 3: RT annotation and Delta-mass are not legible. Please improve for readability.
13. Figure 4: In panel b and c the yellow and orange color are very similar and difficult to distinguish. Consider improving the contrast between the two.
14. LINE 264-265: Please consider rephrasing this sentence for easier understandability.
15. LINE 378: Is there a clinical study number associated with the obtained patient samples? If so, please report this.
16. In figure 5 you mention strong and weak binder, please define somewhere which affinities were associated with either.
17. LINE 838: Figure 6 does not exist in the manuscript version that this reviewer was provided with. Please correct.
18. In the methods section the utilized mass spectrometer for the DDA runs is listed as Q Exactive, while the raw files on PRIDE indicate that a Q Exactive HF-X was actually used. Please correct accordingly.

Reviewer #2 (Remarks to the Author):

Ferreira et al. claim the development of a novel immunopeptidomics workflow that aims to identify circRNA-derived HLA ligands. They validated their workflow using two melanoma cell lines and eight previously published primary lung carcinoma immunopeptidome datasets. The authors critically discuss the current state of art and highlight recent issues in the detection of circRNA-derived HLA ligands which they overcome in their study.

Whereas, I acknowledge the first time detection of circRNA-derived HLA ligands, the biological relevance, and impact of the work is rather limited as for example most data is provided from data reanalysis of published datasets and immunopeptidomes and no functional T cell data is provided. Thus, I fear that the relevance of this work for the field is not high.

Please find below a detailed list of comments and concerns.

Major comments:

- Why does the authors limited their database to the canonical start codon ATG and do not use any alternative translation initiation sites?
- The authors mentioned that their workflow could be used for HLA class II immunopeptidomes but did not show any data on that. It would be very interesting to see if circRNA-derived peptides could also be identified in the HLA class II immunopeptidome and to what extent.
- Why does the authors only focus on the three forward frames?
- The authors stated that they provide a novel workflow. Therefore, it would be important to make the script available for creating the FASTA file as well as the FASTA file itself. The authors stated the FASTA file is uploaded on Pride, which is not the case.
- The authors should focus in their results text and figures (e.g. Sup Fig 1) on the circRNA-derived peptides that overlap the BSJ region otherwise it is an overestimation. Of the 122 circRNA-derived peptide sequences only 17 span the BSJ region! For the IFN-g treated cells the authors directly mentioned the 32 unique circRNA-derived peptides overlapping the BSJ, which should be consistent throughout the manuscript.
- In which intensity range of the immunopeptidome the circRNA-derived ligands are detected? Rank plots could illustrate this.
- It is striking that circRNA-derived ligands are primarily detected for A*68 and A*03. Can this be due to the better detectability by the amino acids K/R at the C-terminus?
- The length distributions and binding % in Sup Fig 6 should be separately depicted for the different conditions. Furthermore, it would be interesting to see the overlap between Spectronout and FragiPipe.
- The DMSO samples seems to be the correct vehicle control for the MG132 treated cells but not for the IFN-g treated cells. However, if DMSO should be used as a control for the IFN-g treated cells the 48 h time point is missing.
- The healthy lung tissues are derived from the same patients? So, it is adjacent benign? This should be clearly stated. It would be better to also analyze true benign tissues as a control.
- The authors stated that 31 circRNA-derived peptides spanning the BSJ region were identified in the lung tissues. However, in Fig 5c 33 are depicted. Maybe it would be helpful to sort the table according to tumor-specific ligand presentation. In Table 2 the HLA annotation is mentioned however it is unclear why

a peptide is annotated as binder and non binder at the same time. Or the same allele is depicted multiple times for the same peptide.

- In their discussion the authors mentioned that one strength of their study in contrast to previous studies is the use of a group-specific FDR. The advantage of such group-specific FDR should be clearly demonstrated in the manuscript. And especially the decision to use 3% FDR instead of the commonly used 1%.
- For circRNA-derived peptides being potential tumor antigens, it is mandatory to investigate their immunogenicity. It would be interesting to know if the circRNA-derived peptides from the BSJ region could elicit T-cell responses. The authors should check for memory T cell responses in lung cancer patients.

Minor comments:

- The Supplement Table file contains some kind of link to another source which should be corrected.
- Line 121: Please shortly explain circBase.
- Table 1: Patient ID isn't the correct term as cell lines are shown. What means infinite translation candidate?
- The authors jump back and forth between the different figures which makes reading difficult.
- It should be stated in the results section that the lung samples are received from previous published studies. It seems that some T1185B samples are already published on Pride (PXD013649). This is not mentioned in the method section.
- Sup Table 1: Why is there no HLA annotation for the peptides with an oxidized methionine?
- Fig 4C: In Sup Table 1 only one peptide is annotated to A*02. This does not match to the pie chart in Fig 4C.
- Sup Fig 7: Are the red marked dots the peptides matching the respective HLA allotype? That is unclear. The circRNA-derived peptides in panel b are not only those matching the BSJ region? It would be better to focus on them.
- It is not clear which mass spec device was used for which samples. Was the DDA performed on the Q Exactive and the DIA on the Eclipse? Why? Doesn't that cause problems in the analysis?
- The authors mentioned specifically the GAST-derived peptides as cancer-related. Did the authors also find canonical GAST-derived peptides as tumor-associated?

Reviewer #3 (Remarks to the Author):

The manuscript presents a workflow to identify circRNA-derived peptides and providing insights into their potential function in tumor immunosurveillance. The identification of hundreds of HLA-bound peptides originating from circRNAs is an exciting finding, and the validation of a group of circRNA-derived peptides through MS adds to the credibility of the study. Several previous studies have reported the identification of many cancer-specific circRNAs. However, it is largely unclear how these circRNAs function in vivo. Here the authors showed that these circRNA can encode protein to expand the peptide sequence library of HLA, which is consistent with our recent discoveries, although our data has not published yet. I appreciate the author provided many validated examples for circRNA RNA and encoded peptides. However, I still have a few concerns and suggestions related to this manuscript.

Major concerns:

1. Regarding the identification of circRNA-derived peptides, it is suggested that the authors could potentially increase the number of identified peptides if they consider using a combined circRNA-derived sequence library from both circBase and corresponding RNA sequencing data. This approach might lead to a more comprehensive representation of circRNA-derived peptides and improve the overall understanding of their immunological relevance.
2. The manuscript mentions the identification of circRNA-derived peptides in three biological replicates of T1185B. It would be beneficial to know the extent of overlap among the identified peptides in these replicates. This information would provide insights into the heterogeneity of HLA expression in the same cell line/tissue and strengthen the reproducibility of the findings.
3. In the MS validation of circRNA-encoded peptides, it is important to consider that the amino acids at both sides of the back-spliced junction should be identified by Mass Spectrometry. I think the author should clarify whether this consideration was taken into account during the validation process, which would provide more confidence in the accuracy of the identified peptides. Otherwise there could be some artifact due to the noise of MS data.

Overall, this manuscript presents a promising study on the identification and potential function of circRNA-derived peptides. Addressing the questions and considering the suggestions mentioned above would further enhance the significance and impact of this work.

REVIEWER COMMENTS

Reviewer #1 (Remarks to the Author):

Humberto Ferreira and colleagues suggest a new methodology and pipeline for the identification of unique circRNA-derived immunopeptides to expand the available pool of tumor-specific antigens and epitopes to further improve anti-cancer immunotherapy. The Bassani-Sternberg lab has previously impressively showcased ways to identify non-canonical immunopeptides from non-coding regions and further their approach with the current work focusing on translated circRNAs. The authors describe how a reference fasta file of circRNA sequences from circBase is created, followed by “stop-to-stop” circRNA fragment identification and in silico translation given the presence of the translation initiation codon ATG upstream of the back-splice junction (BSJ). Peptides were further truncated when applicable to 49 amino acid stretches, in which the central amino acid was at least partially encoded by the BSJ sandwiched by 24 amino acids up- and downstream of the BSJ. The particular lengths were chosen so that also longer MHC class II ligands, with a maximum length of 25 amino acids, might be identified by this approach in the future. The authors use multiple software tools or group-specific FDR to reduce the number of false positive circRNA-derived immunopeptide identifications. They employ targeted MS and Sanger sequencing to confirm their findings and identify potential tumor-specific circRNA-derived immunopeptides with potential for therapeutic intervention.

The manuscript is very well written and provides a very valuable contribution to further expand on the targetable tumor antigenic pool. Two earlier studies have attempted to identify circRNA-specific immunopeptide sequences (as cited by the authors), but due to methodological limitations have not convincingly shown the identifications of immunopeptides unique to translated circRNA. The current manuscript therefore presents the first fully validated attempt to identify MHC class I-presented immunopeptides from translated circRNA. All references included are timely and support the stated points. All relevant data has been uploaded to the PRIDE archive or is already publicly available on PRIDE from previous publications. While the manuscript is of high relevance to the fields of immunopeptidomics, antigen processing and presentation as well as cancer immunotherapy, a number of mostly minor points should be addressed before the manuscript is deemed suitable for publication in the highly esteemed journal Nature Communications.

1. If the reviewer understood the circRNA fasta file generation process correctly, only circRNAs solely composed of exonic sequences are considered for the presented workflow. It would be interesting to know which fraction of all translated circRNAs this corresponds to? If a substantial fraction of translated circRNA is missed by the workflow, the authors might consider discussing a future extension of their pipeline to cover all translated circRNAs.

In silico translation was performed for all the circRNA sequences present in the circBase database, independently on the fact if they were composed solely by exonic sequences or not. Human “Putative spliced circRNA sequences” fasta from circBase contains 140,790 human circRNAs. circBase provides different annotations for each circRNA, including features relative to their genomic location such as “intronic”, “intergenic”, “ncRNA”, “UTR3”, “UTR5”. For illustration purposes, Fig.1a and b refer to circRNAs composed solely by exons, but the strategy was applied to all circRNAs from circBase.

Summarized information was added to the method section and to legend of Fig. 1a: “For illustration purposes, a circRNA composed exclusively by exons is shown and the different back-spliced exons are colored” and Fig. 1b: “Workflow is illustrated using a fictitious transcript sequence derived from an exonic circRNA hosted by a coding gene, but the strategy was applied to all human circRNAs in circBase, independently of their annotation”.

2. An additional figure in the main manuscript about the number and characterization of the identified canonical and circRNA-derived peptides would be appreciated. Even though tables 1 and 2 provide some of this information already, the reviewer believes that it is important to display an overview in peptide length, peptide motifs and binding affinities in the main text to instill confidence into the obtained data, which is available already in the supplements, but should be presented more prominently.

We now refer throughout the manuscript to three groups of peptides:

1. Protein coding (PC)

2. circRNA, that includes all peptides mapping uniquely to circRNA sequences in the fasta,
3. circRNA-BSJ, that includes only peptides that span the back spliced junction.

Figure 2 was updated with the number of identified peptides in T1185B cell line, peptide length distribution, % peptides predicted as binders to the respective HLAs, and the binding restrictions for PC, circRNA, and circRNA-BSJ groups. Graphs were taken from Supp. Fig.1. We did not add clustering of motifs because the BSJ subgroup is too small to visualize the motifs.

Print-screen from Fig 2.

3. The circRNA-derived peptides seem to be slightly longer than the canonical peptides, particularly when focusing on the BSJ-spanning peptide sequences. Please discuss this in the first paragraph of the results or the discussion section. Could these longer peptides trigger a more specific T cell response due to looping out of the binding groove creating a more unique interaction surface for the TCR?

We modified the text as suggested by the reviewer and added some lines in the results section on the potential immunogenicity of longer HLA-I peptides:

"PC derived peptides exhibited the expected length distribution for HLA-I peptides (average length of 9.80), while circRNA, and the subset of circRNA-BSJ peptides were overall longer (average length of 10.07 and 10.11, respectively; Fig. 2b). Indeed, extended HLA-I restricted peptides, longer than 11 amino acids, effectively stimulate CD8⁺ T cell responses within the typical range for epitope-specific CD8⁺ T cells, however, longer, "bulging" peptides may pose challenges for T-cell receptor recognition compared to shorter peptides¹."

4. What could be the reason in your opinion for this interesting preference for a single HLA allele for circRNA-derived peptides? Are strong differences in expression levels the likely cause? What are the expression levels of the respective HLA alleles?

From other analyzes we have performed on T1185B and Mel-1 cells we can exclude loss of HLA heterozygosity, but we found genomic duplications of one chromosome, impacting 3 alleles: HLA-A*68:01, HLA-B*40:01, HLA-C*03:04 in T1185B cells, and HLA-A*03:01, HLA-B*07:02, HLA-C*07:02 in Mel-1 cells. More importantly, in T1185B we found a higher expression at the transcriptomic level of HLA-A*68:01 (as seen in the figure below), which can explain the remarkable representation of this allele to the global peptidome, and consequently also to the fraction of circRNA- and circRNA-BSJ-derived peptides. In addition, we and others have seen a preference for HLA-A*68:01 and HLA-A*03:01 in the non-canonical space which is not yet fully explained. Furthermore, while Lysin and Arginine (K,R) are known to be coded in splicing junctions², we believe that the bias in detection of HLA-A*68:01 and HLA-A*03:01 in these samples is mostly related to the higher expression of these alleles. Since this impacts similarly both PC and circRNAs derived peptides, we decided to not include this additional genomics/transcriptomic data due to the already lengthy manuscript.

Copy number impacting HLA Class-I genes calculated from whole exome sequencing and expression values of HLA genes from RNAseq performed on T1185B and Mel-1 cells.

5. How many of the BSJ-spanning immunopeptides from table 1 were tested in total by PRM and how many of these could not be confirmed? Were the tested peptides that failed PRM QC of rather low search engine score and could potentially be false positives?

For a more transparent interpretation of PRM results, Table 1 was updated with a new classification of the PRM results. We now consider the following:

“+” **validated**: where we confirmed the identification.

“-“ **not validated**: where only the heavy peptides were identified, therefore we could not validate the identification (likely false positives).

“**Failed QC**”: These were due to sticky peptide with very long elution profile. Therefore, we could not validate the identification.

“**Inconclusive**”: Not passing our manual inspection due to noisy data. Therefore, we could not validate the identification.

In total, out of 19 unique circ-RNA-BSJ peptides identified in T1185B we tested by PRM 16 peptides, of which 7 were validated, 4 were non-validated peptides and 5 were inconclusive as they didn't pass our manual inspection due to noisy signal (Table 1).

In Mel-1 we identified 11 unique circ-RNA-BSJ, tested 6 and validated 4 (Table 1). Here, one peptide failed QC due to too long elution profile.

This information was added to Table 1 and to the method section:

“Following manual inspection of the results, peptides were annotated as PRM-validated (+) and PRM non-validated (-). Other tested peptides were defined as ‘failed QC’ (due to too long elution profile), and as ‘inconclusive’ due to noisy signal (Table 1).”

For your information, we here provide the PRM results of the inconclusive and failed QC:

Inconclusive and failed QC PRM results. Skyline elution profiles of peptides (a) STMSTNGIPRGR, (b) STDLFWTVK, (c) QTPFAFHPR, (d) NTASMAGGFLLR, (e) TIYEESFFR, and (f) KLLHGKLVFK. Heavy-labeled amino acids of the synthetic peptides are shown in bold.

6. For this reviewer, it is hard to confirm from the current Volcano plots in panels 4f and 4h that the circRNA-derived BSJ-spanning peptides follow the same trend upon IFN γ and MG132 treatment as the canonical presented immunopeptides. Please consider a different graphical representation of the global up- and down-regulation of canonical and circRNA-derived immunopeptides to show this more conclusively and potentially even allow statistical testing.

We thank the reviewer for this comment. We changed the visualization of the volcano plots and added new plots in Figure 4, demonstrating the difference in the fold-change of peptide intensities in the two treatments, per HLA allele.

Violin plots added to panels (g) and (h) of main Figure 4 of the manuscript. (g) Volcano plot representing the differential presentation of circRNA-derived peptides (separately for those covering the junction and those that do not) upon IFN γ treatment (left) and an associated violin plot showing the difference in the log₂ intensity for the peptides according to their HLA restriction (right). Due to the overrepresentation of HLA-A*68:01 in circRNA and circRNA-BSJ groups, only this HLA allele is represented for circRNAs. There was no significant difference observed in the presentation of peptides (diff. log₂ intensity) by comparing the PC group and the circRNA-BSJ subgroup restricted to the mentioned allele. (h) Same as g, for the MG132 treatment. In the volcano plots, circRNA-no-BSJ peptides and circRNA-BSJ are highlighted in blue and red, respectively.

7. LINE 80: Please add a brief sentence to explain the back-splice junction since some readers might not be familiar with it.

The generation of the back-splicing junction (BSJ) is first described and explained in the second paragraph of the introduction (before line 80):

“In humans, such non-polyadenylated transcripts are produced by a non-canonical splicing process, known as back-splicing, between two non-sequential exons where the 3' end of a downstream exon is fused to the 5' end of an upstream exon³. The generated junction is called back-splicing junction (BSJ).”

8. LINE 128: Add here the lengths of MHC class I and II peptides for readers to more easily see why 49 amino acids were chosen as peptide truncation length.

This information was added to the manuscript.

“HLA-I peptides are short (8-15 amino acids, mostly 9 mers), while HLA-II peptides may reach up to 25 amino acids (average length of around 15-16 amino acids). Therefore, where possible, sequences were further trimmed to a length of with up to 49 amino acids covering the transcript position corresponding to at least one BSJ (24 amino acids upstream the BSJ, one amino acid partially encoded by the BSJ and 24 amino acids downstream the BSJ). This made the circRNA-derived BSJ-ORF fasta reference suitable for both HLA-I and HLA-II MS-based immunopeptidomics workflows (Fig. 1b; see Methods section).”

9. Tables 1 and 2: Please change the color-code of the peptide sequence and the 5' exon to facilitate readability/ease of information intake for red-green colorblind people.

Color-code was changed in Table 1, Table 2 and Suppl. Table 1 from green and red to black and orange to facilitate readability. Also, heatmaps from new Supp. Fig. 7 (e) and bubble plots from Fig. 4 (d), (e) and (f) have now a colorblind friendly color-code.

10. Which peptides were found in both the DDA and DIA experiments? Please include a column to indicate this information in Table S1.

Sixteen peptide sequences out of nineteen from Table 1 are common between the two MS searches. Corresponding column was added to Table S1.

11. Figure 2: The annotations of b and y-ions are not legible in the spectra and the sequence above seems to be distorted. Please correct.

We recreated the spectra, now the heavy and light peptides are mirrored for a better interpretation. Size of the annotations of the b and y-ions was increased for a better readability as well.

12. Figure 3: RT annotation and Delta-mass are not legible. Please improve for readability.

Size of the RT and Delta-mass annotations was increased for a better visualization.

13. Figure 4: In panel b and c the yellow and orange color are very similar and difficult to distinguish. Consider improving the contrast between the two.

Darker orange color was used to increase the contrast in Fig. 4 b and c. For harmonization, same color pattern was changed across the other figures, when applicable.

14. LINE 264-265: Please consider rephrasing this sentence for easier understandability.

This sentence was rephrased:

“To properly control false identifications and to properly account for lower likelihood of true identification of non-canonical sequences⁵⁶, we implemented a group-specific FDR control in FragPipe⁵⁷⁻⁶⁰. When calculating the FDR, canonical and non-canonical peptides were classified into different groups. The FDR was calculated for each group separately because different groups have different score distributions. Here, we applied a group-specific FDR of 3%, allowing stringent control of error in the circRNA search space.”

15. LINE 378: Is there a clinical study number associated with the obtained patient samples? If so, please report this.

We used published data for the lung cancer cohort that is not from a specific clinical study. Mel-1 is the only sample specifically used from study NCT03475134 and this is now reported in the methods section: Patient samples, cell lines and cell culture.

16. In figure 5 you mention strong and weak binder, please define somewhere which affinities were associated with either.

This part of the figure was replaced to highlight another circRNA-BSJ peptide as the previous one did not pass another filter we introduced to answer a comment from reviewer 3.

17. LINE 838: Figure 6 does not exist in the manuscript version that this reviewer was provided with. Please correct.

This was corrected, it corresponded to Figure 5.

18. In the methods section the utilized mass spectrometer for the DDA runs is listed as Q Exactive, while the raw files on PRIDE indicate that a Q Exactive HF-X was actually used. Please correct accordingly.

We corrected the text in the methods, accordingly. Data generated for this project was acquired by either a Q Exactive HF-X (DDA runs) or an Orbitrap Eclipse (DIA runs), while data taken from PRIDE accession PXD013649 and PXD034772 was acquired by a Q Exactive HF and a Q Exactive HF-X, respectively. The mass spectrometer instrument used to acquire each raw file is now mentioned in Supplementary Table 7.

Reviewer #2 (Remarks to the Author):

Ferreira et al. claim the development of a novel immunopeptidomics workflow that aims to identify

circRNA-derived HLA ligands. They validated their workflow using two melanoma cell lines and eight previously published primary lung carcinoma immunopeptidome datasets. The authors critically discuss the current state of art and highlight recent issues in the detection of circRNA-derived HLA ligands which they overcome in their study.

Whereas, I acknowledge the first time detection of circRNA-derived HLA ligands, the biological relevance, and impact of the work is rather limited as for example most data is provided from data reanalysis of published datasets and immunopeptidomes and no functional T cell data is provided. Thus, I fear that the relevance of this work for the field is not high.

Please find below a detailed list of comments and concerns.

Major comments:

- Why does the authors limited their database to the canonical start codon ATG and do not use any alternative translation initiation sites?

We don't exclude the possibility that circRNA translation can be also driven by alternative translation initiation sites, but to enhance database efficiency and robustness, we restricted it to the canonical start codon ATG. This choice was made because ATG initiation sites exhibit a higher translation efficiency, making the occurrence of these non-canonical circRNA translation events more probable.

- The authors mentioned that their workflow could be used for HLA class II immunopeptidomes but did not show any data on that. It would be very interesting to see if circRNA-derived peptides could also be identified in the HLA class II immunopeptidome and to what extent.

To explore circRNA-BSJ peptides in the HLA-II immunopeptidome, we analyzed the matched HLA-II immunopeptidome of the lung cancer cohort. We performed DIA-DDA analysis with FragPipe, enabling the group-specific 3% FDR threshold and added this analysis to Supplementary Figure 9. Only two circRNA-BSJ peptides were identified, and this fits well with previous studies that demonstrated a very low detection rate of non-canonical peptides in the HLA-II immunopeptidome ^{4, 5}. Search results were uploaded to PRIDE.

- Why does the authors only focus on the three forward frames?

Normally, one would use 6 frames translation if the direction of the transcription of a gene is unknown. Here, this information is available from the annotated gene transcript in the circBase.

- The authors stated that they provide a novel workflow. Therefore, it would be important to make the script available for creating the FASTA file as well as the FASTA file itself. The authors stated the FASTA file is uploaded on Pride, which is not the case.

FASTA files were already provided in PRIDE in the first submission (in the zipped 'search' folders). The script to generate the FASTA file and to modify it to align with FragPipe's group-specific FDR calculation is now available in GitHub:

"The concatenated fasta file was adapted for compatibility with the group-specific FDR calculation in FragPipe following the structure of UniProt fasta file headers⁶. Two groups were assigned through the attribution of a different number in the Protein Existence field of the fasta headers (PE). PE=1 was attributed to the protein group (including common MS protein contaminants) and PE=4 was attributed to the circRNA group, allowing FragPipe to annotate sequences to the different groups for the group-specific FDR calculation. The script used to produce the circRNA-derived BSJ-ORF fasta reference is available on the following GitHub link: https://github.com/bassanilab/CircRNA_MS_ref_fasta."

- The authors should focus in their results text and figures (e.g. Sup Fig 1) on the circRNA-derived peptides that overlap the BSJ region otherwise it is an overestimation. Of the 122 circRNA-derived peptide sequences only 17 span the BSJ region! For the IFN-g treated cells the authors directly

mentioned the 32 unique circRNA-derived peptides overlapping the BSJ, which should be consistent throughout the manuscript.

We thank the reviewer for raising this concern. We now refer to 'circRNA-BSJ' (covering the junction), 'circRNA-not-BSJ' (all other circRNA-derived peptides), and 'circRNA' derived peptides' when we refer to both groups together. This is now fixed throughout the text and figures.

"The NewAnce tool⁷ was used to calculate a group-specific FDR of 3%, for both MaxQuant and Comet, for peptides derived from the human UniProt entries (namely group protein-coding 'PC') and peptides derived uniquely from the circRNA-derived BSJ-ORF fasta reference (group 'circRNA'). Only peptides identified by both search engines were retained. In this study, we were particularly interested in characterizing circRNA-derived peptides that spans the BSJ, named 'circRNA-BSJ'. Importantly, our searches against the circRNA group resulted in identification of other circRNA-derived peptides that do not overlap the BSJ (specifically named below as 'circRNA-not-BSJ' when applicable), that could potentially derive from out-of-frame translation events of the matched linear RNAs (with canonical or alternative translational initiation sites), which also represent a potential source of non-canonical antigens. Their inclusion in the study was not within the study's intended scope. They were added for the sake of comparison and completeness."

- In which intensity range of the immunopeptidome the circRNA-derived ligands are detected? Rank plots could illustrate this.

Overall, the circRNA-BSJ peptides span the entire range of intensities. Rank plots of peptide intensities were added in Figure 2 for T1185B, and Supp. Figure 2 for Mel-1.

Print-screen of Figure 2.

Print-screen of Supplementary Figure 2.

- It is striking that circRNA-derived ligands are primarily detected for A*68 and A*03. Can this be due to the better detectability by the amino acids K/R at the C-terminus?

From other analyzes we have performed on T1185B and Mel-1 cells we can exclude loss of HLA heterozygosity, but we found genomic duplications of one chromosome, impacting 3 alleles: HLA-

A*68:01, HLA-B*40:01, HLA-C*03:04 in T1185B cells, and HLA-A*03:01, HLA-B*07:02, HLA-C*07:02 in Mel-1 cells. More importantly, in T1185B we found higher expression at the transcriptomic level of HLA-A*68:01 (as seen in the figure below), which can explain the remarkable representation of this allele to the global peptidome, and consequently also to the fraction of circRNA- and circRNA-BSJ-derived peptides. In addition, we and others have seen a preference for HLA-A*68:01 and HLA-A*03:01 in the non-canonical space which is not yet fully explained. Furthermore, while Lysin and Arginine (K,R) are known to be coded in splicing junctions², we believe that the bias in detection of HLA-A*68:01 and HLA-A*03:01 in these samples is mostly related to the higher expression of these alleles. Since these impacts similarly both PC and circRNAs derived peptides, we decided to not include this additional genomics/transcriptomic data due to the already lengthy manuscript.

Copy number impacting HLA Class-I genes calculated from whole exome sequencing and expression values of HLA genes from RNAseq performed on T1185B and Mel-1 cells.

- The length distributions and binding % in Sup Fig 6 should be separately depicted for the different conditions. Furthermore, it would be interesting to see the overlap between Spectronaut and Fragpipe.

As we generated new immunopeptidomics data (IFN γ versus control, addressing your comment below) we re-analyzed all the data again with Spectronaut (using same parameters as before) and with FragPipe, but this time using a 3% group-specific FDR, aiming for consistency with the subsequent lung cohort analyses. We included a comparison of the overlap between the two search engines in terms of unique peptide sequences detected in the PC, circRNA and circRNA-BSJ groups (Suppl. Fig. 7a). We also show the number of unique peptide sequences with predicted HLA restriction per replicate and per condition detected by each of the search engines (Suppl. Fig. 7b). Moreover, the percentage of predicted binders is now shown per replicate and per condition for the two search engines individually and for the intersection of the two (Suppl. Fig. 7c). Length distributions are now shown per replicate and per condition for PC, circRNA and circRNA-BSJ groups, for peptides identified by both FragPipe and Spectronaut (Suppl. Fig. 7d).

- The DMSO samples seems to be the correct vehicle control for the MG132 treated cells but not for the IFN-g treated cells. However, if DMSO should be used as a control for the IFN-g treated cells the 48 h time point is missing.

We thank the reviewer for this important note. We repeated the experiment in order to include the correct control for IFN γ . We treated T1185B cells with IFN γ , or control (water), for 48hr. We collected three biological replicates of 20 million cells for each condition, performed the affinity purification of HLA-I complexes, and performed DIA MS measurements of eluted peptides. To increase the coverage of the spectral library, we assembled available DDA raw files from T1185B cells treated or not with IFN γ (from the PRIDE accession PXD0136497) and those we included in the current manuscript), together with the DIA files of T1185B cells treated with MG132 and the new IFN γ treatments (and their respective controls). These files were processed with the hybrid-DIA analyses processes with FragPipe group-specific 3% FDR and with Spectronaut with 1% FDR. Overall, we observed similar results about the presentation of the circRNA-BSJ derived peptides. We included new plots to visualize the lack of significant differences in presentation of circRNA-BSJ and PC

derived peptides following the treatments. These are now included in the revised manuscript and in Figure 4.

New scheme summarizing the treatment experiments.

Violin plots added to panels (g) and (h) of main Figure 4 of the manuscript. (g) Volcano plot representing the differential presentation of circRNA-derived peptides (separately for those covering the junction and those that do not) upon IFN γ treatment (left) and an associated violin plot showing the log₂ intensity for the peptides according to their HLA restriction (right). Due to the overrepresentation of HLA-A*68:01 in circRNA and circRNA-BSJ groups, only this HLA allele is represented for circRNAs. There was no significant difference observed in the presentation of peptides (diff. log₂ intensity) by comparing the PC group and the circRNA-BSJ subgroup restricted to the mentioned allele. (h) Same as g, for the MG132 treatment. In the volcano plots, circRNA-no-BSJ peptides and circRNA-BSJ are highlighted in blue and red, respectively.

- The healthy lung tissues are derived from the same patients? So, it is adjacent benign? This should be clearly stated. It would be better to also analyze true benign tissues as a control.

The reviewer is right, they correspond to adjacent benign healthy lung tissues from same patient as the tumor. Importantly, to address a comment from reviewer 3, we included another filtering step to increase the robustness of our analysis and to remove potentially ambiguous identifications resulting from PSMs that better fit possible modified PC sequences. The MSMS spectra of candidate circRNA-BSJ peptides were re-searched with COMET against the human reference proteome UniProt database concatenated with the list of the circRNA-BSJ peptide sequences, including six common modifications. Ambiguous identifications mapping to modified PC peptides with either higher or equal XCorr (delta score =0) were excluded from downstream analyses. Due to this stringent filter, the peptide mapping to GAST circRNA was removed from the manuscript. Instead, we highlighted the peptide ILDKKVE[KL] in the revised manuscript. This peptide is predicted to be encoded by the circRNA hsa_circ_0076651 which is another example of an ORF with putative infinite translation. Interestingly, the peptide ILDKKVEKL was detected in five of the lung cancer patients, exclusively in

tumor tissues, and it could not be identified in any benign tissue included in the HLA Ligand Atlas⁸. We included the following sentences in the revised manuscript:

"Therefore, we leveraged large HLA-I and HLA-II immunopeptidomics DIA and DDA datasets of tumoral and adjacent healthy matched multi-region tissues from eight lung cancer patients and searched for circRNA-BSJ derived peptides presented specifically, or to a higher extent, in the tumors."

"Fifteen of these circRNA-BSJ peptides were predicted to bind any of the HLA-I molecules expressed in at least one of the patients where they were detected (Supplementary Table 4). Although none of these were deemed cancer-related based on their host gene expression in TCGA/GTEX (see Methods), six circRNA-BSJ peptides were uniquely detected in the tumor tissues, including the peptide ILDKKVE[KL]. This peptide is predicted to be encoded by the circRNA hsa_circ_0076651 which is another example of an ORF with putative infinite translation. Interestingly, the peptide ILDKKVEKL was detected in five of the lung cancer patients, exclusively in tumor tissues, and it was not identified in any benign tissue included in the HLA Ligand Atlas⁸."

- The authors stated that 31 circRNA-derived peptides spanning the BSJ region were identified in the lung tissues. However, in Fig 5c 33 are depicted. Maybe it would be helpful to sort the table according to tumor-specific ligand presentation. In Table 2 the HLA annotation is mentioned however it is unclear why a peptide is annotated as binder and non binder at the same time. Or the same allele is depicted multiple times for the same peptide.

We apologies for the confusion. Initially, two peptides with Methionine oxidation were counted twice (modified and non-modified). Now we removed this duplication, having 31 unique sequences in total in table 2. However, in the heatmap of Figure 5c we maintained 33 peptides (31 unique sequences) since we wanted to discriminate in which patients and tissue type (healthy or tumor) the oxidized and non-oxidized forms of ASMAGGFLLR and DAALIKMV peptides were detected.

In the previous version of Table 2, we included information on the peptides as binders and non-binders in any of the patients, ending up with a peptide being a binder in one patient and a non-binder in another. However, we agree with the reviewer that this is confusing, and we now included a new Supp. Table 4 with the predicted HLA restriction (or annotating the peptides as non-binders: "NB"), adding the corresponding predicted affinity in % rank values, in each lung patient.

- In their discussion the authors mentioned that one strength of their study in contrast to previous studies is the use of a group-specific FDR. The advantage of such group-specific FDR should be clearly demonstrated in the manuscript. And especially the decision to use 3% FDR instead of the commonly used 1%.

The importance and advantage of using group-specific FDR in proteogenomic applications have been already discussed extensively in the literature⁹. In addition, specifically for non-canonical immunopeptidomics, we have already published a comprehensive analysis demonstrating the importance of stringently controlling false identification with group-specific FDR as well as the importance of considering the intersection of two search engine tools⁷. Therefore, we include a citation to that earlier study, and due to the lengthy text, we prefer not to repeat same investigation here as well. Our results in T1185B cell line, post IFN γ treatment or proteasome inhibition with MG132, clearly demonstrate the improved accuracy of the identifications, comparing the percentage of circRNA-derived peptides predicted as binders in FragPipe 3% group-specific FDR (90.1%), Spectronaut 1% global FDR (87.8%), and the intersection of the two tools (96%).

We decided to use 3% group specific FDR following our earlier study (Chong et al.)⁷ where we have reasoned that it gives a good compromise of sensitivity while keeping overall high accuracy (as estimated by the fraction of peptides predicted as binders). This is arbitrary, and commonly in immunopeptidomics an FDR of 1-5% is used. If results are validated with complementary assays, we believe this is sufficiently justified.

- For circRNA-derived peptides being potential tumor antigens, it is mandatory to investigate their immunogenicity. It would be interesting to know if the circRNA-derived peptides from the BSJ region

could elicit T-cell responses. The authors should check for memory T cell responses in lung cancer patients.

We agree with the reviewer that experimentally validating the immunogenicity of circRNA-BSJ would be informative for translational research and future clinical application, unfortunately, we do not have viable T cells from these lung cancer patients. Nevertheless, we anticipate that our study will stimulate further investigation into this unique source of peptides, elucidating their clinical implications.

Minor comments:

- The Supplement Table file contains some kind of link to another source which should be corrected.

This has been corrected.

- Line 121: Please shortly explain circBase.

A brief description of circBase was added: "circBase is the first repository of circRNAs which merged different datasets of circRNAs, offering an interface with standardized annotations and unique identifiers".

- Table 1: Patient ID isn't the correct term as cell lines are shown. What means infinite translation candidate?

Patient ID was changed to Sample name.

In the introduction, we elucidate that circRNAs with open reading frames (ORFs) lacking stop codons can engage in endless translation through a process known as rolling circle translation.

"Efficient circRNA translation has also been demonstrated in studies using exogenous circRNAs with infinite open reading frames (ORFs) lacking stop codons, undergoing rolling circle translation¹⁰."

- The authors jump back and forth between the different figures which makes reading difficult. We have corrected this by reordering the figures.

- It should be stated in the results section that the lung samples are received from previous published studies. It seems that some T1185B samples are already published on Pride (PXD013649). This is not mentioned in the method section.

This has been corrected in the revised manuscript:

"To increase the coverage of the spectral library, we assembled available DDA raw files from T1185B cells treated or not with IFN γ (from the PRIDE accession PXD013649⁷), the newly generated DDA data, together with the DIA files of T1185B cells treated with MG132 and the new IFN γ treatments (and their respective controls), as indicated in Supplementary Table 7. "

- Sup Table 1: Why is there no HLA annotation for the peptides with an oxidized methionine?

This has been corrected in the revised manuscript.

- Fig 4C: In Sup Table 1 only one peptide is annotated to A*02. This does not match to the pie chart in Fig 4C.

We thank the reviewer for noticing this. It was a mistake that we corrected in the revised manuscript.

- Sup Fig 7: Are the red marked dots the peptides matching the respective HLA allotype? That is unclear. The circRNA-derived peptides in panel b are not only those matching the BSJ region? It would be better to focus on them.

Indeed, the red dots were related to peptides mapping to the HLA allotypes. We now changed the headers to improve clarity, now in Supp Figure 8. In main Figure 4, the color clearly indicates the circRNA-BSJ or circRNA-not-BSJ.

For example (from new Supplementary Figure 8):

- It is not clear which mass spec device was used for which samples. Was the DDA performed on the Q Exactive and the DIA on the Eclipse? Why? Doesn't that cause problems in the analysis?

In this study MS data were acquired by either a Q Exactive HF-X (DDA runs) or a Orbitrap Eclipse (DIA runs), while data taken from PRIDE accession PXD013649 and PXD034772 were acquired by a Q Exactive HF and a Q Exactive HF-X, respectively. The mass spectrometer instrument used to acquire each raw file was also added to Supplementary Table 7.

Specifically, MS searches related to the T1185B treatments with MG132 and IFN γ were performed using a hybrid approach. Using this method a comprehensive library was generated, based on DDA files from 3 biological replicates of the untreated cell line acquired using the Q Exactive HF-X, publicly available DDA files from the same cell line treated with IFN γ (known to boost the number presented peptides) acquired with a Q Exactive HF, and the newly generated DIA files of the MG132 and IFN γ treatments, obtained with an Orbitrap Eclipse. The generated library was used to match the DIA files for peptide identification and quantification purposes. Combining such data is not problematic for identification purposes. Importantly, the differential presentation analyses were performed for HLA peptide eluted from low input samples, and the samples were measured sequentially using the same acquisition method and same LC-MS instrument and setups (Orbitrap Eclipse). Therefore, in this case, the DIA acquisition that was done with the newer generation of instrument is expected to improve the detection of circRNA-derived peptides.

- The authors mentioned specifically the GAST-derived peptides as cancer-related. Did the authors also find canonical GAST-derived peptides as tumor-associated?

As described above, to address a comment from reviewer 3, we included another filtering step to increase the robustness of our analysis and to remove potentially ambiguous identifications resulting from PSMs that better fit possible modified PC sequences. Due to this stringent filter, the peptide mapping to GAST circRNA was removed from the manuscript.

Instead, we highlighted in Figure 5 the peptide ILDKKVE[KL] that is predicted to be encoded by the circRNA hsa_circ_0076651 which is another example of an ORF with putative infinite translation, hosted by the *HSP90AB1* gene. Interestingly, the peptide ILDKKVEKL was detected in 5 five of the lung cancer patients, exclusively in tumor tissues, and it was not identified it in any benign tissue included in the HLA Ligand Atlas⁸.

Reviewer #3 (Remarks to the Author):

The manuscript presents a workflow to identify circRNA-derived peptides and providing insights into their potential function in tumor immunosurveillance. The identification of hundreds of HLA-bound peptides originating from circRNAs is an exciting finding, and the validation of a group of circRNA-derived peptides through MS adds to the credibility of the study. Several previous studies have reported the identification of many cancer-specific circRNAs. However, it is largely unclear how these circRNAs function in vivo. Here the authors showed that these circRNA can encode protein to expand the peptide sequence library of HLA, which is consistent with our recent discoveries, although our data has not published yet. I appreciate the author provided many validated examples for circRNA RNA and encoded peptides. However, I still have a few concerns and suggestions related to this manuscript.

Major concerns:

1. Regarding the identification of circRNA-derived peptides, it is suggested that the authors could potentially increase the number of identified peptides if they consider using a combined circRNA-derived sequence library from both circBase and corresponding RNA sequencing data. This approach might lead to a more comprehensive representation of circRNA-derived peptides and improve the overall understanding of their immunological relevance.

We thank the reviewer for this interesting comment. To test if indeed adding circRNA events captured by sample-specific RNAseq data could improve the sensitivity of the peptide detection we generated a sample specific circRNA reference from total RNAseq data of Mel-1, with the circRNA discovery pipeline described in Memczak et al 2013¹¹. Total RNAseq data was prepared using the Illumina TruSeq Stranded Total RNA reagents according to the protocol supplied by the manufacturer and sequenced on a HiSeq 4000, resulting in 2 x 88,567,719 paired-end 151bp RNA-seq reads. We downloaded the latest version of the 'find_circ' pipeline (v1.2) from the GitHub repository (https://github.com/marvin-jens/find_circ) and applied it to the Mel-1 RNAseq data. The RNA-seq reads were aligned to the GRCh37 reference assembly, and the 58,763,750 unmapped reads (33.2% of the input) were processed by the find_circ pipeline to generate 4,180 filtered (high quality) circRNA candidates. Based on the chromosomal coordinates, 3,195 (76.4%) of these candidate circRNAs were already present in the circBase database. Of the remaining 985 candidates, 765 (18.3%) could be assigned Ensembl transcript annotation using GENCODE (v38lift37), leaving 220 candidates (5.3%) that were likely intergenic. We then generated a Mel-1 specific fasta reference based on circRNA events found in Mel-1 sample, excluding the likely intergenic circRNAs.

We compared search results of FragPipe group-specific 3%FDR of Mel-1 immunopeptidomic data, using the following references: CircBase, circBase+Mel-1, and Mel-1 only, each concatenated to UniProt. While we identified several circRNA-BSJ peptides, none mapped uniquely to the Mel-1 fasta (table below). This could be related to the difficulty in identifying BSJ reads in RNAseq data even in a sample with a remarkable good overall read depth as Mel-1. However, we should carefully acknowledge that this is a preliminary analysis, performed on a single sample. We concluded from this that the generic large circBase database is comprehensive enough and provides a decent reference to capture relevant candidate peptides, and therefore the chances of finding novel peptides, or to significantly improve the detection of more peptides is actually not high. We therefore adjusted the text in the discussion to now emphasize that novel circRNAs not already listed in circRNA databases could potentially lead to discovery of novel peptides.

“Newly discovered circRNA sequences obtained through capture sequencing¹² and nanopore sequencing¹³ could be used as input for the generation of a sample-specific circRNA reference file to potentially identify sample-specific circRNA-derived peptides.”

Gene ID	circRNAs frame	Infinite translation candidate	Peptide Sequence	Length (AA)	NetMHCpan - 4.1 Binder/Best Allele	BSJ context (3'Exon -> Peptide [BSJ] <- 5'Exon)	Mel-1 FASTA entry	circBase	circBase+Mel1	Mel-1
ABCA2	hsa_circ_0089621_2	FALSE	RLKEPSTQR	9	HLA-A03:01	ISWVYSVAMTIQHIVAEKEH->RLK[EP]STQR<-RP	FALSE			
ATP1A1	hsa_circ_0013693_1	FALSE	IRRRPGGTR	9	no_binder	AFPYSLLIIFYDEVKRLI->IRRRPG[G]TR<-S	FALSE			
BABAM1	hsa_circ_0049967_0	FALSE	TFNYLPGPV	9	no_binder	GLTSDPRELCSCLYDLETASCS->TF[N]YLPGPV<-RGNVTAKAGVVQRLQNQR	FALSE			
BZW2	hsa_circ_0079491_0	FALSE	KIMKPSETMLR	11	HLA-A03:01	M->KIMKPSETML[R]<-	FALSE			
CCAR1	hsa_circ_0018553_1	FALSE	VLSKGGKPPK	9	HLA-A03:01	ISAASITPLLOTPOPLLQOPOQK->[V]LSKGGKPPK<-	TRUE			
CEP57L1	hsa_circ_0130265_1	FALSE	KLRDPTDSTLR	11	HLA-A03:01	MGLSSCKN->KLRDPTDSTL[R]<-AWYLVQT	FALSE			
CSRP2BP	hsa_circ_0059519_1	FALSE	KIKGKRRLP	9	no_binder	LVHNKPTMKPEGEKLSASTL->KIK[G]KRRLP<-GGAPVQVWASAWVEVPTSVO	FALSE			
CTNNB1	hsa_circ_0004030_1	FALSE	RVFEVYHTTVLK	12	HLA-A03:01	SWMGCLQVTAISWPLILTKSSF->[R]VFEVYHTTVLK<-IQRGOWLLKLI	FALSE			
	hsa_circ_0003137_1	FALSE				MATKKA->[R]VFEVYHTTVLK<-IQRGOWLLKLI	FALSE			
DOCK4	hsa_circ_0133049_1	TRUE	NSIVIITTV	9	HLA-A02:01	LIDKLDLMSSEKGDYRELF->NS[I]VIITTV<-LYLSDALRKNFLNENFDY	FALSE			
EIF4G1	hsa_circ_0068342_1	FALSE	SPRSPSTSL	10	HLA-B07:02	VAGTQFPASAKVAAPLTPHD->SPRSP[S]JSTL<-AGPSRAVQPPPECRVQPLPA	FALSE			
EIF4G3	hsa_circ_0010464_1	FALSE	NAQIAITVP	9	no_binder	MSS->[A]QIAITVP<-KTWKKPKDRTRTTEEML	FALSE			
	hsa_circ_0010455_1	FALSE				MQT->[A]QIAITVP<-KTWKKPKDRTRTTEEML	FALSE			
GRB10	hsa_circ_0134422_0	FALSE	SVIVKIGR	8	no_binder	NSVALGAPLCSRRVLYLRARPPQS->[S]VIVKIGR<-HLYKLWHCV	FALSE			
IFT81	hsa_circ_0097243_0	FALSE	AEQENLGKLK	10	HLA-B44:02	QQEKRAIREQYTKNT->AEQENLG[KL]K<-L	FALSE			
KLHL29	hsa_circ_0119386_1	FALSE	ALRLRSPPR	9	HLA-A03:01	LPAPPAARPPASWP->ALRLRSP[PR]<-LSGFLVMLTLC	FALSE			
MALAT1	hsa_circ_0096124_2	FALSE	KLLHGKVVFK	11	HLA-A03:01	M->[KL]LHGKVVFK<-RKLRETTPEINT	FALSE			
MYPN	hsa_circ_0018492_1	FALSE	VRGTVKAR	9	no_binder	SGCFTCTASNYGTVSSIAQLH->VR[G]GTVKAR<-SLKIPQIFTSSRGEICTH	FALSE			
NFKB1	hsa_circ_0070531_1	FALSE	TSLPLSSKL	9	HLA-C05:01	MIC->T[S]LPLSSKL<-QSKILLONOPCLCS	FALSE			
None	hsa_circ_0003418_1	FALSE	SPRSLRSGP	9	HLA-B07:02	MTSVSTMIPTLWMSAWTPTA->[S]PRSLRSGP<-LKYSISA	FALSE			

PATL1	hsa_circ_0022208_0	FALSE	GAVVFGGL	8	no_binder	FGGLGEEDEEIQFNDDTFFGS->GAV[V]FGGL<-SSG	FALSE	
PDI3	hsa_circ_0035034_2	FALSE	RVSSYFVLH	9	HLA-A03:01	LLERFLLLSSELLKERSLSCRSS->[R]VSSYFVLH<-ISLTSRLRLWHIQSK	FALSE	
PPP2R3A	hsa_circ_0122039_2	FALSE	KVLSLFTEK	9	HLA-A03:01	SPVGDKAKDITSAVLIQQTPEVI->[KV]SLSFTEK<	FALSE	
REPS2	hsa_circ_0139997_0	FALSE	TVATKSGLLPPP	12	no_binder	MVQV->TV[A]TKSGLLPPP<-PALPPRCPQSQSEQV	FALSE	
SAE1	hsa_circ_0051662_2	FALSE	ILAGEIVKV	9	HLA-A02:01	MAPVCAVVG->ILAGEIV[KV]L<-LLLRDGPVCGGWDRDFGTGNCESG	FALSE	
SPTAN1	hsa_circ_0088876_2	FALSE	MMTSPKAGLL	10	no_binder	->MMTSPK[AG]LL<-KKHEAFETDFTVHKDRVNDVC	FALSE	
TOPBP1	hsa_circ_0121989_1	FALSE	KSLAAELLVLK	11	HLA-A03:01	MYTPHCART->K[S]LAAELLVLK<	FALSE	
ZMYM4	hsa_circ_0113154_2	FALSE	RVVSWIQK	9	HLA-A03:01	DKAANQVEETLHHLPTPETNF->[RV]VSWIQK<-CLKI	FALSE	
ZNF512B	hsa_circ_0115585_0	FALSE	QISSLLGK	8	no_binder	SSLRARSYSFSDSFTAAS->QISS[L]LGK<-SS	FALSE	

2. The manuscript mentions the identification of circRNA-derived peptides in three biological replicates of T1185B. It would be beneficial to know the extent of overlap among the identified peptides in these replicates. This information would provide insights into the heterogeneity of HLA expression in the same cell line/tissue and strengthen the reproducibility of the findings.

The overlap in biological replicated in peptides identified in PC, circRNA, and circRNA-BSJ is shown in Figure 2 for T1185B where 31% of circRNA-BSJ peptides were consistently detected in all three biological replicates. This percentage is within the range observed for PC (35%) and all peptides derived from circRNA (32%). In addition, in the lung cancer cohort, the detection of circRNA-BSJ derived peptides in each of the healthy or tumor regions is now visible in Supp Figure 9, for both HLA-I and HLA-II immunopeptidomes.

Print-screen of Figure 2 demonstrating reproducibility of detecting peptides in T1185B samples.

3. In the MS validation of circRNA-encoded peptides, it is important to consider that the amino acids at both sides of the back-spliced junction should be identified by Mass Spectrometry. I think the author should clarify whether this consideration was taken into account during the validation process, which would provide more confidence in the accuracy of the identified peptides. Otherwise there could be some artifact due to the noise of MS data.

We thank the reviewer for his concern about the correct identification of circRNA-encoded peptides. Due to the inherent ambiguity in peptide spectrum matches and the higher probability of having false identification in DDA and DIA MS/MS in the non-canonical space, we initially implemented a stringent group-specific FDR control and considered circRNA-BSJ peptides that were identified by two search engines. In addition, in the revised manuscript we included another filtering step to increase the robustness of our analysis and to remove potentially ambiguous identifications resulting from PSMs that better fit possible modified PC sequences. The MSMS spectra of candidate circRNA-BSJ peptides were re-searched with COMET against the human reference proteome UniProt database concatenated with the list of the circRNA-BSJ peptide sequences, including six common variable modifications. Ambiguous identifications mapping to modified PC peptides with either higher or equal XCorr (delta score = 0) were excluded from downstream analyses.

In addition, our strategy for validation included the synthesis of heavy labeled counterparts and their spike-in into newly generated samples for targeted validation by LC-MS (PRM). Indeed, in some cases (included in Supp Figure 4), the PRM fragmentation pattern provides validation where fragment ions do support the presence of each of the amino acids in the junction, for example, in the case of [TT]ENIPVRR circRNA-BSJ peptide shown below. However, it is important to clarify that co-elution of endogenous and heavy labeled transitions confirms unconditionally the presence of circRNA-

encoded peptide sequence in the sample as well as the correctness of amino acid at both sides of the back-spliced junction, regardless of the location of the junction in the peptide. Confirmation of circRNA-encoded peptides through this PRM approach will effectively eliminate false identifications originating from noisy MS data. Obviously, this validation approach is very expensive and time consuming, therefore the stringent FDR control is expected to minimize false identification, yet, it is unavoidable that some identifications might still be false.

Overall, this manuscript presents a promising study on the identification and potential function of circRNA-derived peptides. Addressing the questions and considering the suggestions mentioned above would further enhance the significance and impact of this work.

We express our gratitude to all the reviewers for dedicating their time to reviewing our manuscript and offering valuable and constructive feedback.

1. Josephs, T.M., Grant, E.J. & Gras, S. Molecular challenges imposed by MHC-I restricted long epitopes on T cell immunity. *Biol Chem* **398**, 1027-1036 (2017).
2. Wang, X. et al. Detection of Proteome Diversity Resulted from Alternative Splicing is Limited by Trypsin Cleavage Specificity. *Molecular & cellular proteomics : MCP* **17**, 422-430 (2018).
3. Nigro, J.M. et al. Scrambled exons. *Cell* **64**, 607-613 (1991).
4. Prensner, J.R. et al. What Can Ribo-Seq, Immunopeptidomics, and Proteomics Tell Us About the Noncanonical Proteome? *Molecular & cellular proteomics : MCP* **22**, 100631 (2023).
5. Kraemer, A.I. et al. The immunopeptidome landscape associated with T cell infiltration, inflammation and immune editing in lung cancer. *Nat Cancer* **4**, 608-628 (2023).
6. UniProt, C. UniProt: the Universal Protein Knowledgebase in 2023. *Nucleic Acids Res* **51**, D523-D531 (2023).
7. Chong, C. et al. Integrated proteogenomic deep sequencing and analytics accurately identify non-canonical peptides in tumor immunopeptidomes. *Nature communications* **11**, 1293 (2020).
8. Marcu, A. et al. HLA Ligand Atlas: a benign reference of HLA-presented peptides to improve T-cell-based cancer immunotherapy. *j. immunotherapy cancer* **9** (2021).
9. Nesvizhskii, A.I. Proteogenomics: concepts, applications and computational strategies. *Nature methods* **11**, 1114-1125 (2014).
10. Abe, N. et al. Rolling Circle Translation of Circular RNA in Living Human Cells. *Sci Rep* **5**, 16435 (2015).
11. Memczak, S. et al. Circular RNAs are a large class of animal RNAs with regulatory potency. *Nature* **495**, 333-338 (2013).
12. Vo, J.N. et al. The Landscape of Circular RNA in Cancer. *Cell* **176**, 869-881 e813 (2019).
13. Zhang, J. et al. Comprehensive profiling of circular RNAs with nanopore sequencing and CIRI-long. *Nat Biotechnol* **39**, 836-845 (2021).

REVIEWERS' COMMENTS

Reviewer #1 (Remarks to the Author):

All my comments and requests have been fully addressed by the authors and I therefore highly recommend publication of this excellent work.

Reviewer #2 (Remarks to the Author):

The authors have addressed all points raised.

Reviewer #3 (Remarks to the Author):

This is the revised version of a previous manuscript. The authors addressed my comments pretty well, and the revised manuscript is much improved than the original version. I recommend for publication.